# Two-stage opening of the Dover Strait and the origin of island Britain

Sanjeev Gupta[1], Jenny S. Collier[1], David Garcia-Moreno[2,3], Francesca Oggioni[1], Alain Trentesaux[4], Kris Vanneste[2], Marc De Batist[3], Thierry Camelbeeck[2], Graeme Potter[5], Brigitte Van Vliet-Lanoë[6] & John C.R. Arthur[7]

Late Quaternary separation of Britain from mainland Europe is considered to be a consequence of spillover of a large proglacial lake in the Southern North Sea basin. Lake spillover is inferred to have caused breaching of a rock ridge at the Dover Strait, although this hypothesis remains untested. Here we show that opening of the Strait involved at least two major episodes of erosion. Sub-bottom records reveal a remarkable set of sediment-infilled depressions that are deeply incised into bedrock that we interpret as giant plunge pools. These support a model of initial erosion of the Dover Strait by lake overspill, plunge pool erosion by waterfalls and subsequent dam breaching. Cross-cutting of these landforms by a prominent bedrock-eroded valley that is characterized by features associated with catastrophic flooding indicates final breaching of the Strait by high-magnitude flows. These events set-up conditions for island Britain during sea-level highstands and caused large-scale re-routing of NW European drainage.

[1] Department of Earth Science and Engineering, Imperial College, South Kensington Campus, Exhibition Road, London SW7 2AZ, UK. [2] Service of Seismology and Gravimetry, Royal Observatory of Belgium, Ringlaan, 3, Brussels B-1180, Belgium. [3] Department of Geology, Renard Centre of Marine Geology, Ghent University, Krijgslaan 281s.8, Ghent B-9000, Belgium. [4] UMR 8187 LOG CNRS/UFR des Sciences de la Terre, Université de Lille 1, Villeneuve d'Ascq, Plouzané, Lille F-59650, France. [5] 5 Snowberry Court, Warwick Gardens, Taunton, Somerset TA1 2PZ, UK. [6] UMR CNRS 6538 CNRS Géosciences Océan, Université de Bretagne occidentale, IUEM, Plouzané, F-29280, France. [7] Top-Hole Site Studies Ltd, Richmond, Surrey TW9 1UW, UK. Correspondence and requests for materials should be addressed to S.G. (email: s.gupta@imperial.ac.uk).

The geographic insularity of Britain from continental Europe is a consequence of high interglacial sea levels that led to marine flooding of the shallow shelf areas of the English Channel and North Sea[1]. Prior to the opening of the Dover Strait, however, Britain remained connected to Europe, even during sea-level highstands, via a structural ridge that extended from southeast England to northwest France. This ridge, made of chalk, comprised the northern limb of the Weald–Artois anticline, and is postulated to have formed a narrow isthmus separating marine embayments to the north (North Sea) and southwest (English Channel)[2,3]. Breaching of this barrier is a necessary prerequisite to form island Britain.

It is widely considered that the breaching of the chalk ridge and opening of the Dover Strait is a consequence of spillover of a proglacial lake that occupied the present-day Southern North Sea basin during the Marine Isotope Stage (MIS) 12 glaciation (∼450 ka and conventionally equated to the Elsterian–Anglian glaciation)[3–7]. Coalescence of the British and Scandinavian ice sheets in the northern and central North Sea basin and the chalk barrier to the south caused fluvial discharge from European rivers and glacial meltwater to become ponded to form this lake[2,6]. Lake spillover is thought to have released a significant megaflood into a subaerial English Channel[4,5]. An extensive network of bedrock-eroded valleys[4,8–10] in the central English Channel shows morphologies characteristic of erosion by high magnitude flood flows[5,7], and has been interpreted as a consequence of catastrophic drainage of a pro-glacial lake[5,7]. Others contend that while a proglacial lake existed in the southern North Sea, its spillover was not a catastrophic process[11]. An alternative model

suggests that breaching of the Strait was an incremental process with slow erosion of the Weald-Artois rock barrier at the Dover Strait by fluvial processes during glacial sea-level lowstands and tidal erosion during interglacial highstands[12,13]. In this model, there is no requirement for a proglacial lake in the southern North Sea basin. Thus models for erosion of the palaeovalley networks downstream of the Dover Strait range from those proposing erosion by high-magnitude events[5,7] to those suggesting relatively slow fluvial erosion[13]. Testing of models for opening of the Strait has, however, been limited through lack of high-resolution marine geophysical data at the inferred breach point in the Dover Strait.

Early marine geophysical investigations in the centre of the Strait in support of the Channel Tunnel engineering project in the 1960s and 1970s identified a set of enigmatic sediment-infilled depressions, termed the Fosse Dangeard (fosse being 'deep' in English)[14,15]. At the time of their discovery, glacial erosion by an icestream advancing through the Dover Strait was proposed to explain the formation of both the bedrock valley and associated depressions[14,16], but this was subsequently discounted due to a lack of independent evidence for the presence of ice this far south[17] (Supplementary Fig. 1). Smith[4], in the first proposal of the megaflood hypothesis, speculated that the depressions might represent fossil plunge pools formed at the base of waterfalls overspilling the chalk barrier from a southern North Sea proglacial lake. Erosion of the Lobourg Channel was inferred to represent the final stage in the breaching of the rock dam. The lack of modern high-resolution marine geophysical data in the Dover Strait has meant that the lake overspill and dam breaching model for opening of the Strait is untested.

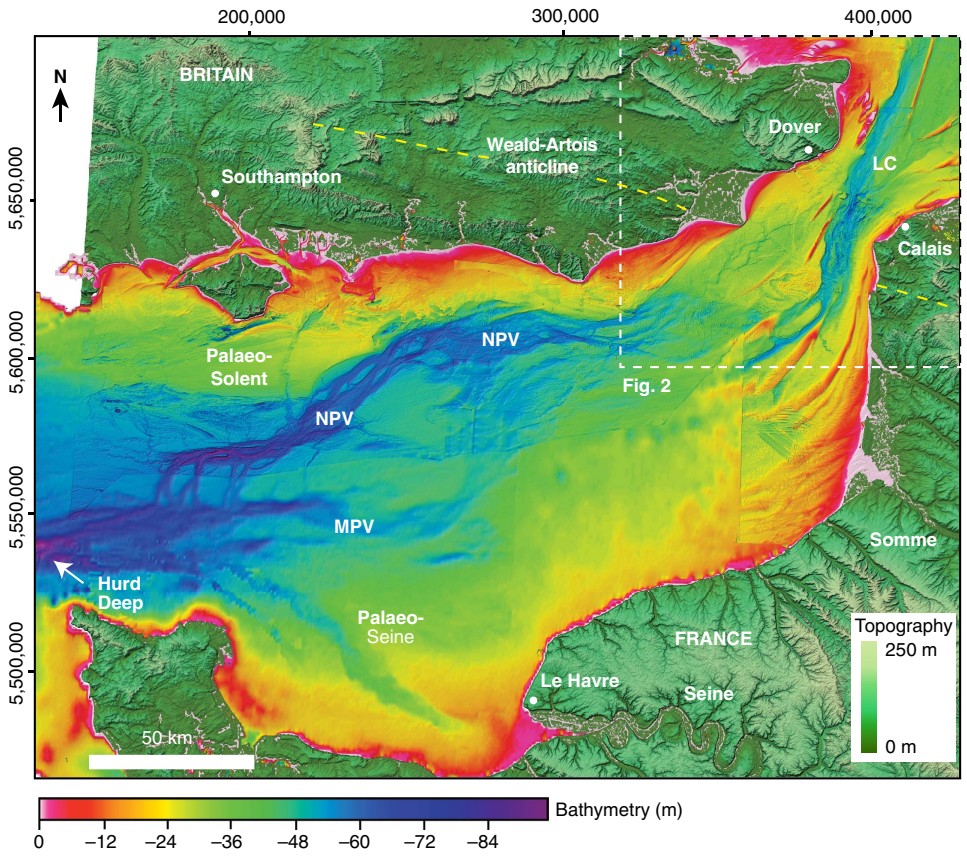

**Figure 1 | Sonar bathymetry of the eastern English Channel shelf.** Map showing bathymetry of the eastern English Channel gridded at 30 m cell size. The location of the study area in Fig. 2 is indicated. LC, Lobourg Channel; MPV, Median Palaeovalley; NPV, Northern Palaeovalley. An analysis of the downstream morphology is given in ref. 7. See Methods for details of offshore bathymetry data. Onshore elevation is from SRTM[51]. Yellow dashed line indicates the axial trace of the Weald–Artois anticline.

Here we combine analysis of a regional high-resolution bathymetric grid of the central and eastern English Channel shelf that includes new multibeam sonar data from the Dover Strait, with new high-resolution seismic-reflection data to investigate how the Strait was formed. The mechanism and history of the breaching of the Dover Strait is a question of importance to not only understanding the geographic isolation of Britain from continental Europe[3], but also the large-scale re-routing of northwest European drainage[11,18,19] and meltwater to the North Atlantic via the Channel[20–23]. Moreover, the opening of the Strait has significance for the biogeography[1,24–26] and archaeology of NW Europe, with particular attention on the pattern of early human colonization of Britain[11,27–29].

## Results

**Seabed geomorphology in the English Channel.** A regional-scale compilation of bathymetric data from the central and eastern English Channel (Fig. 1) shows that a prominent valley, the Lobourg Channel[7,13,30], in the Dover Strait can be traced southwestwards into a network of bedrock-eroded valleys in the central Channel[5,7]. Individual valleys, such as the Northern Palaeovalley, have an anabranching planform morphology with channels bifurcating around streamlined bedrock remnants[5,7]. The continuity of the valley network, in particular the Northern Palaeovalley, with the Lobourg Channel in the Dover Strait establishes them as part of the same palaeodrainage system[7] (Fig. 2). For example, the Lobourg Channel at its southwestern extent bifurcates around a prominent bedrock streamlined island (SI 1) to form two channel pathways. Because the valleys in the central Channel are interpreted to have formed by erosion by high-magnitude flood flows[5,7], their continuity with the Lobourg Channel suggests that incision of this prominent feature was likely a related process. Below we examine the Fosse Dangeard depressions and the high-resolution morphology of the Lobourg Channel to explore their processes of formation within the context of the English Channel palaeodrainage system.

**Subsurface geomorphology of the Dover Strait.** To investigate the early landscape evolution of the Strait, we used high-resolution seismic profiles to describe the morphology of the Fosses Dangeard depressions in the central part of the Dover Strait (Methods; Supplementary Fig. 2). Our lines cover the entire Strait and so allow us to map the detailed distribution, geometry and internal architecture of the Fosses Dangeard. These data show that the Fosses consist of a series of kilometre–diameter depressions, the base of which are incised up to ~140 m into Cretaceous bedrock (Figs 3a and 4). This depth of incision is exceptional given the low gradient setting of the English Channel shelf. Mapping of the Fosses in adjacent seismic lines shows that the deepest parts of the depressions are spatially discontinuous though connected by shallow corridors. In total, we recognize seven major depressions, which we label A–G (Fig. 3b). Seismic facies in the Fosses infill vary greatly from one Fosse to the next making correlation of seismic units impossible without groundtruthing.

The cross-sectional geometry of the Fosses Dangeard is illustrated in the seismic reflection profile in Fig. 3a. Here the basal erosion surface in both Fosses A and B describes a concave-up cross-sectional profile in a NE–SW orientation. Fosse A is an elongate, WNW–ESE-oriented depression that has a length of ~4 km and width of 0.9 km. The depression has a depth of ~80 m with flank slopes of up to 15°. The depression is eroded into Cretaceous strata that form the hanging wall of a reverse fault, and progressively cuts through the Lower Chalk and Gault

Clay into the Lower Greensand strata (Fig. 3a). The Fosse sediment infill comprises two distinct seismic units. The lower-most unit is characterized by a transparent seismic facies with weak parallel internal reflectors that drape and onlap the basal erosion surface. This lower infill is truncated by a distinct erosional surface (labelled IES) that indicates a second episode of scouring, before continued infilling. Fosse B is a ~2 km radius depression with a sub-circular planform that is eroded ~80 m through Lower Greensand sandstones into strata of the Wealden Beds (Fig. 3a). The stratigraphy of this Fosse also comprises two distinct seismic units separated by a prominent internal erosion surface indicating that Fosse infilling was not continuous.

The spatially localized occurrence of the Fosses depressions is clearly demonstrated in Fig. 4, which shows the geometry of Fosses D and E in consecutive NE–SW-oriented seismic profiles. Fosse E shows marked variability in both width and depth of erosion when traced from the northwest to southeast. By contrast, Fosse D is a ~90 m deep depression with a concave-up geometry in three-dimensions (Fig. 5). Sub-horizontal strata within Fosse D onlap the basal erosion surface on all its flanks indicating it forms a closed depression.

**Spatial distribution of the Fosses Dangeard.** A map of sediment thicknesses within the Fosses (Fig. 3b) reveals that they form spatially isolated sub-circular to elliptical features clustered in a ~7-km-wide, WNW–ESE-oriented belt, perpendicular to the Strait between Dover and Calais (Fig. 6). This belt is parallel to the strike of Cretaceous strata occurring immediately south-west of the boundary between Lower Cretaceous bedrock and the Lower Chalk. Similar deep, bedrock-eroded, sediment-infilled depressions have not been reported either to the northeast or southwest of this narrow belt in the Strait. A fluvial valley interpretation is not tenable as they form closed three-dimensional (basin-shaped) erosional features, rather than two-dimensional (valley-shaped) features. Sets of the Fosses not only occur along the width of much of the Strait, but also occur in a NE–SW orientation parallel to the Strait, for example, Fosses A, B and C.

**Bedrock geology and structural influence on Fosses geometry.** The role of bedrock geology and pre-existing structure is an important consideration in characterizing the geometry of the Fosses Dangeard. The Fosses are eroded into a variety of lower Cretaceous bedrock lithologies with varying resistance to erosion. Fosses D and E are eroded up to 80 m into relatively more resistant Lower Greensand and Lower Chalk strata (Fig. 4), whereas Fosses B, C and F are eroded into the less resistant Weald Clay (Fig. 3). In the latter case, we find that the Fosses are relatively deeper and with a greater planform diameter than those eroded into more resistant lithologies. Thus, while the formation of the Fosses is not purely localized by differential erosion along weaker bedrock strata, bedrock lithology does influence their overall geometry. Similarly, pre-existing structures influence but do not govern Fosses formation. Fosse A is eroded along a WNW–ESE-oriented fault zone[31,32], which it cross-cuts and which constrains its geometry as indicated by its WNW–ESE-oriented elongate geometry. By contrast, Fosse D is eroded into horizontally stratified bedrock and is not associated with any structures (Figs 4 and 5). Similarly, a seismic profile across Fosse B shows that while its north-eastern margin is eroded into the flank of a fault-controlled monocline, the overall geometry of Fosse B is not structurally controlled (Fig. 3)[32]. We conclude that pre-existing structures, although influencing Fosses geometry, are not a primary control on their formation.

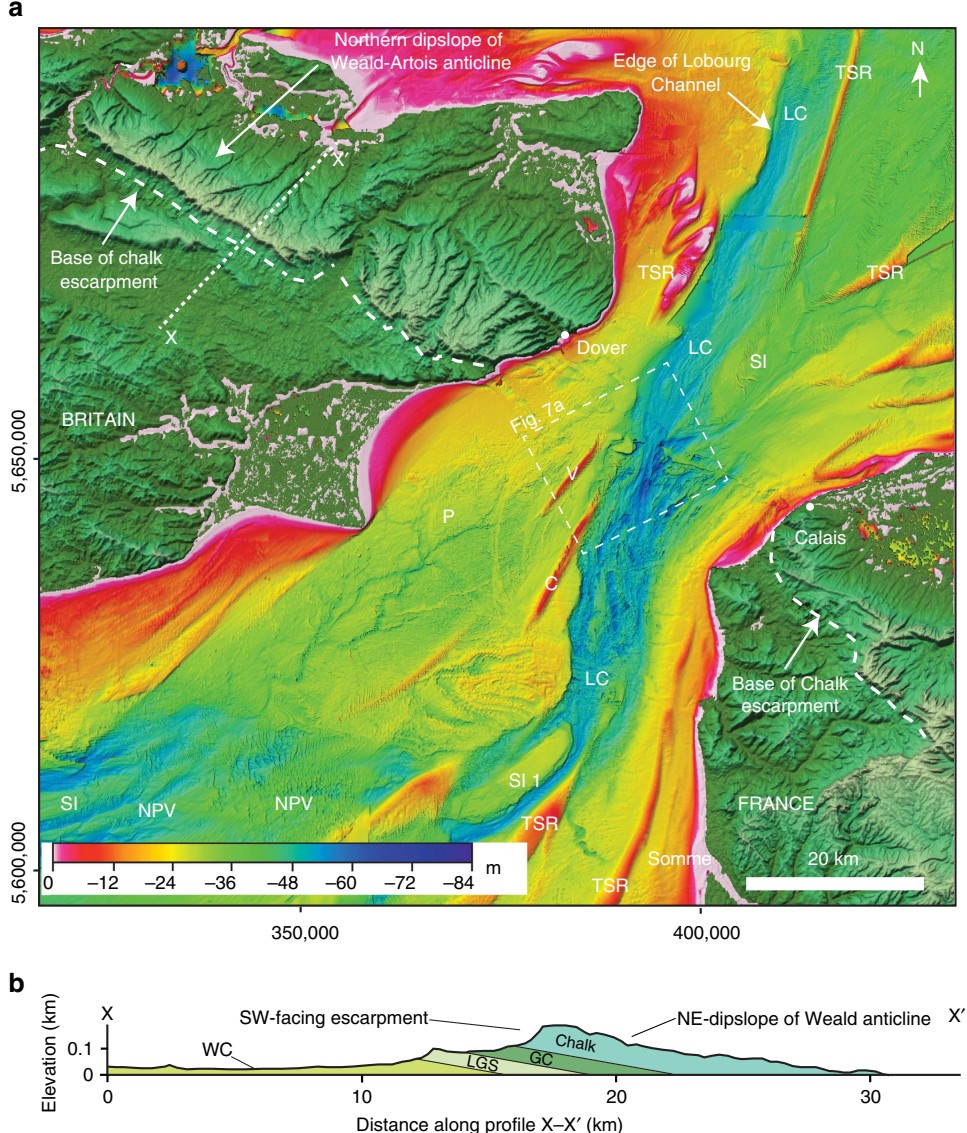

**Figure 2 | Sonar bathymetry of the northeastern English Channel shelf.** (**a**) Coloured and shaded relief bathymetry compilation map of the Dover Strait region. C, Colbart tidal sand ridge; LC, Lobourg Channel; NPV, Northern Palaeovalley; P, platform; V, Varne tidal sand ridge; SI, streamlined island; TSR, tidal sand ridge. Onshore topography (scale in Fig. 1) shown as coloured shaded relief is from Shuttle Radar Topography Mission (SRTM)[51]. Note valley network incised into platform, P, indicating sub-aerial exposure of platform surface. Water depth is indicated by colour bar. X–X′, line of geological cross-section in **b** is indicated. (**b**) Geological cross-section across northern flank of Weald anticline onshore. Note the prominent SW-facing Chalk escarpment and NE-facing gentle dip slope. GC, Gault Clay; LGS, Lower Greensand; WC, Weald Clay.

**Geomorphic interpretation of the Fosses Dangeard**. The plan-view and cross-sectional geometry of the Fosse depressions, together with their great depth of erosion and spatially localized occurrence leads us to interpret them as giant plunge pools, as originally speculated by Smith[4]. We propose that they formed by vertical drilling into bedrock by water jets derived from large waterfalls[33–35]. Similar features are described albeit on a much smaller scale in natural examples[33,35,36] and in laboratory experiments[37]. The remarkable depth of erosion of these features into bedrock suggests that there must have been a substantial elevation drop adjacent to the scours to generate water jets capable of such erosion. It is also plausible that the plunge pools were formed as bedrock step-pool features by high-magnitude water flow down the steep SW-facing slope of the escarpment[38,39]. It is difficult to explain these giant depressions by either normal fluvial, glacial[14,16] or tidal[40] erosion processes.

The plunge pool depressions are aligned with the offshore extension of a southwest-facing Chalk escarpment formed by the northeastern limb of the eroded Weald–Artois anticline (Figs 2 and 6). The topographic profile of this escarpment onshore is markedly asymmetric with a shallow NE-facing dipslope and a steep, SW-facing escarpment (Fig. 2b). Since this topographic feature is inferred to have formed the structural dam that impounded the southern North Sea proglacial lake at its southern limit[3–6], the presence of deeply eroded depressions at the foot of the escarpment strongly supports a model in which the plunge pools were formed by water cascading over this structural dam as large waterfalls. Lake water trapped to the northeast of the escarpment likely overspilled it to form large waterfalls that excavated the plunge pools. The escarpment retreated such that in the final stage the plunge pools eroded chalk bedrock. In this model, escarpment retreat by overspill would have produced sequential plunge pool erosion from Fosses C to A (Fig. 3b).

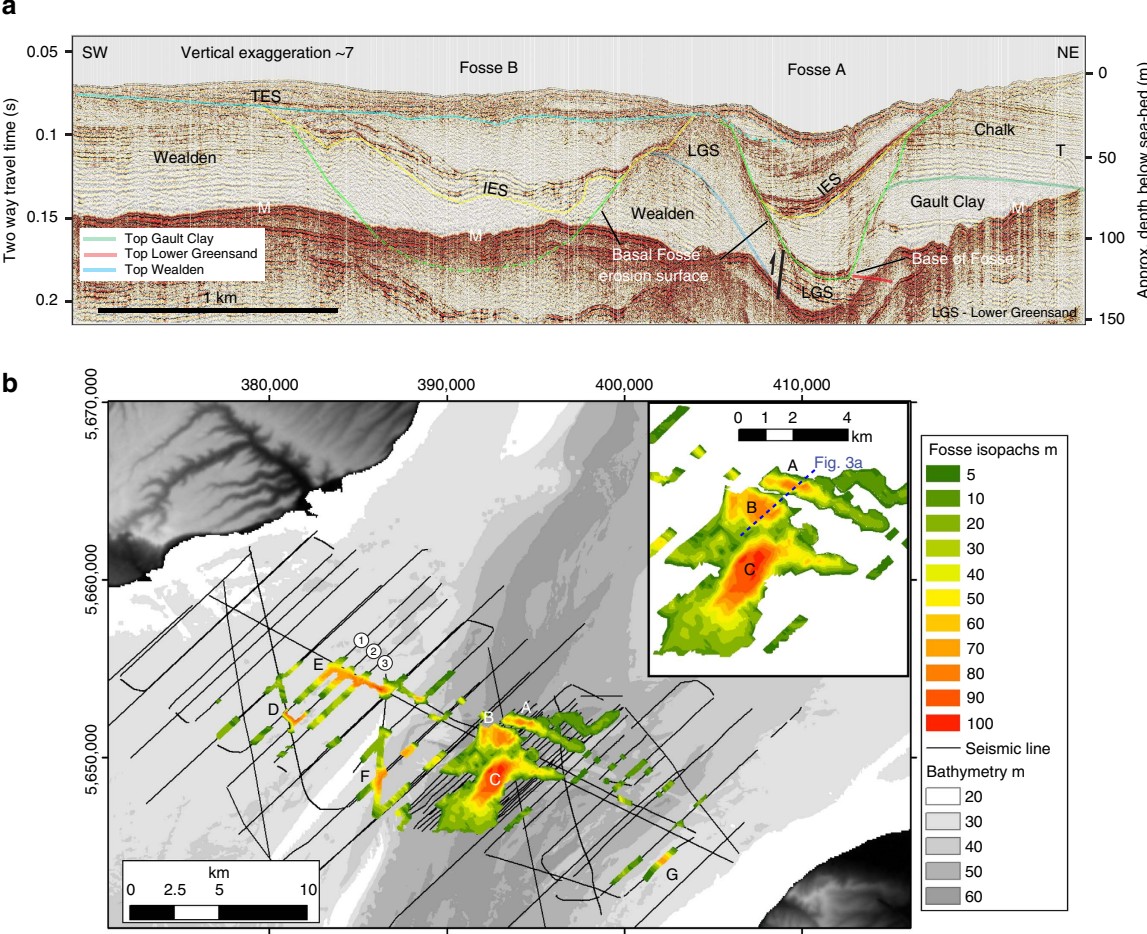

**Figure 3 | Morphology and distribution of the Fosses Dangeard depressions. (a)** Interpreted seismic reflection profile across Fosses Dangeard depressions A and B showing cross-sectional geometry of Fosses and bedrock geology. IES, internal erosion surface within Fosse sedimentary infill; M, seabed multiple; T, seismic diffraction pattern caused by Channel Tunnel; TES, transgressive erosion surface. Location of seismic profile indicated in **b** inset. **(b)** Map showing sediment thickness in Fosses Dangeard depressions. Inset shows detail of Fosses A, B and C. Note absence of colour along seismic tracklines indicates bedrock exposed at seabed indicating that the depressions are highly localized.

Similarly, in the northwestern part of the Strait, Fosse D may have been eroded when the escarpment was located at its northeastern margin. Subsequent, retreat of the escarpment ~4 km to the northeast led to erosion of Fosse E. This relative timing of Fosses erosion cannot be constrained by our data. The infilling history of the Fosses remains poorly understood without lithological calibration. However, the seismic data reveal that after their excavation the plunge pools experienced multiple episodes of infilling and scouring. This likely relates to the complex multi-stage history of the area as it was repeatedly exposed and transgressed during subsequent glacial and interglacial cycles.

**Seabed geomorphology of the Dover Strait.** Our new compilation of marine geophysical data shows that the erosion and infilling of plunge pools at the base of the rock dam was not the only event that shaped the Dover Strait. A second event is interpreted on the basis of erosional landforms identified from analysis of high-resolution multibeam bathymetric data (~2-m-grid-spacing) coupled with regional single-beam bathymetric data (Methods; Supplementary Fig. 2).

The data allow a new interpretation of the Lobourg Channel, which extends through the centre of the Strait from the southern North Sea basin into the English Channel (Fig. 2). The

compilation shows the Lobourg Channel to be a ~80-km-long and ~10-km-wide NE–SW-oriented valley. It comprises a ~25-m-deep valley eroded into Cretaceous bedrock, with a box-shaped cross-sectional profile (Fig. 7a). Locally, lozenge-shaped bedrock remnants with streamlined margins are observed that we interpret as streamlined islands (Fig. 7a), as identified in the downstream extension of the Lobourg Channel, the Northern Palaeovalley in the central English Channel[5,7].

The floor of the Lobourg Channel is itself locally incised by two narrow inner channels, ~500 m wide, that are linear to slightly sinuous in planform (Fig. 7a,b). Channel 1 is >3.5 km long and terminates to the northeast in an amphitheatre-shaped head; it shows relief of ~15 m at the seafloor. It is eroded into relatively resistant sandstones of the Lower Greensand Formation. Channel 2 shows ~20 m of relief at the seabed, is ~200–500 m wide and branches to the northeast into two amphitheatre-shaped heads. The inner channel headwall reaches a maximum of 24 m height, ~1 km wide and has a slope of ~10°–15° (Fig. 7b). Seismic data reveal the channel and headwall are incised into bedrock, with the headwall formed of relatively resistant sandstones of the Lower Greensand Formation and the channel base in weaker claystones at the base of the Lower Greensand (Fig. 8). A prominent erosional scour occurs in an alcove at the base of the headwall (Fig. 8), which bathymetry data indicates is ~10 m deep. Immediately north of the headwall, a ~600-m-wide ovoid scour

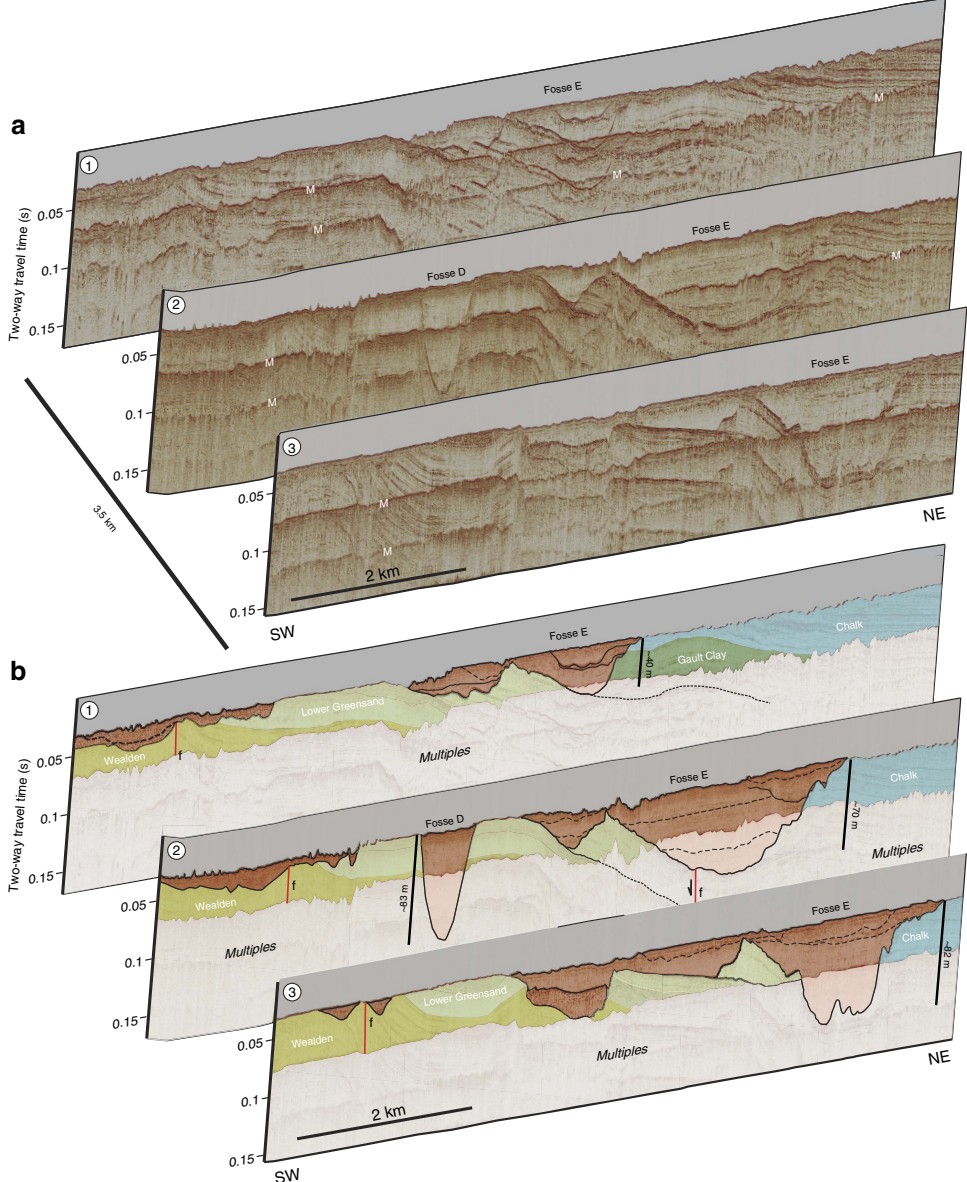

**Figure 4 | Spatial variability in geometry of Fosses Dangeard depressions.** (**a**) Three-dimensional perspective view of seismic profiles across north-west sector of Dover Strait study area showing geometry of depressions D and E. Depressions form localized features eroded in Cretaceous bedrock. Note presence of possible internal erosion surfaces within Fosses sediment infill. Vertical exaggeration ~13. M, seabed multiple. (**b**) Geological interpretation of seismic profiles with details of Cretaceous bedrock into which Fosses are eroded. Locations of seismic profiles indicated in Fig. 3b. f, fault.

is eroded ~20 m into Lower Greensand bedrock (Fig. 7b). The floor of the Lobourg Channel truncates sediment-infilled Fosses Dangeard depressions that underlie it, indicating that erosion of the Lobourg Channel post-dates Fosses incision and infilling (Fig. 8).

We interpret the amphitheatre-shaped channel heads as abandoned cataracts formed by focussed knickpoint propagation similar to those formed by high-magnitude floods in the Channeled Scabland[41,42], Snake River Plains[43] and Iceland[44,45]. Upstream migration of cataracts during flood flow carved the narrow, bedrock inner channels downstream of the amphitheatre heads. The 10-m-deep erosional scour at the base of the headwall of Channel 2 is suggestive of plunge pool erosion by water plunging over the cataract lip (Figs 7b and 8). Similar plunge pools occur in alcoves at the base of cataracts in the Channeled Scabland[42] and flood-eroded terrains in the Snake River Plains of Idaho[43]. Upstream retreat of the cataract to form Channel 2 may

have been facilitated by the strong-over-weak bedrock stratigraphy and toppling failure in the Lower Greensand sandstone lithology[46]. The depth of erosion into bedrock at the seafloor associated with the cataracts is evidence of powerful erosional processes, although it should be noted that the magnitude of these is much smaller than the Fosses Dangeard depressions that we interpret as plunge pools.

Upstream of the inner channels, we discovered several sets of highly parallel streamlined ridge-and-groove bedforms eroded into Chalk bedrock on the floor and immediate flank of the Lobourg Channel (Fig. 7c; Supplementary Figs 3 and 4). These linear bedforms have a consistent ENE–WSW alignment (060°–240°), sub-parallel to the axis of the Lobourg Channel (Supplementary Fig. 3). Individual lineations are up to ~30 m wide, ~5 km in length, and have amplitudes of 0.5–1.2 m (Supplementary Fig. 5). Locally sand dunes lie orthogonally across these features indicating that the lineations are not in

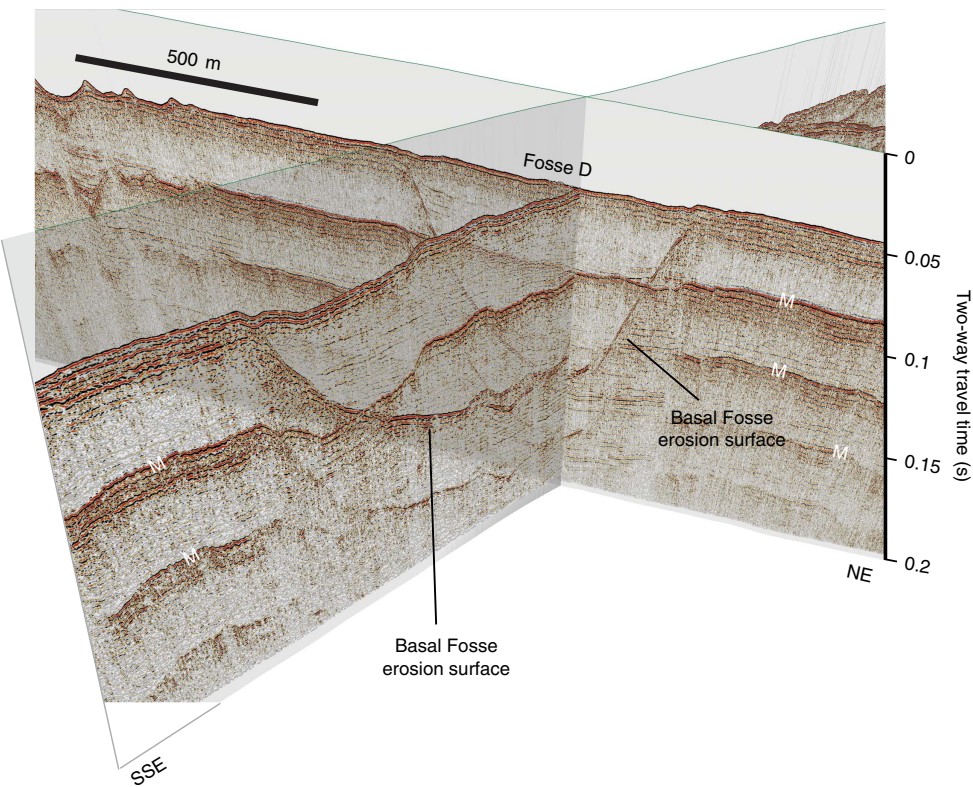

**Figure 5 | Three-dimensional perspective view of seismic profiles across Fosse D.** The NNW–SSE-oriented seismic line has been made partially transparent to view the geometry in NE–SW-oriented line. Note how the basal erosion surface of the Fosse, which is cut into Lower Cretaceous bedrock, shows a bowl-shaped geometry in cross-cutting seismic profiles. Vertical exaggeration ∼4. M, seabed multiple. Locations of seismic profiles are indicated in Supplementary Fig. 2.

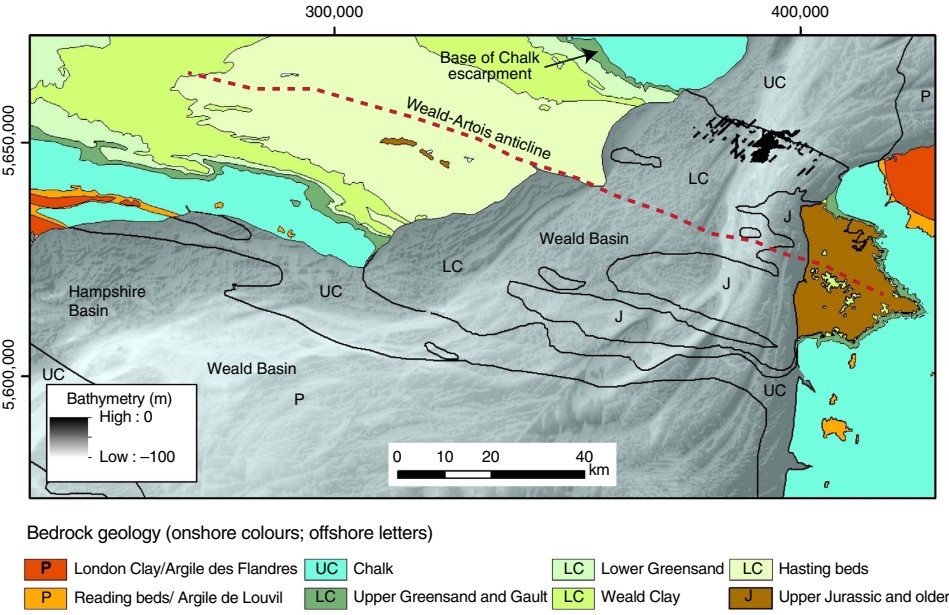

**Figure 6 | Map showing onshore and offshore bedrock geology of Dover Strait area.** Onshore bedrock geology shown in colours and offshore geology bedrock geology indicated by letters. The zone of black colour in Strait indicates locations of sediment-infilled Fosses Dangeard along seismic tracklines. Note how the Fosses are localized immediately southwest of the Chalk bedrock outcrop in the northern sector of Strait. This correlates onshore with the base of the SW-facing escarpment formed by the Chalk. LC, Lower Cretaceous, UC, Upper Cretaceous, P, Palaeogene.

equilibrium with the current flow field in the Strait. The origin of these highly parallel linear erosional bedforms remains enigmatic. Our preferred interpretation is that these represent bedrock-eroded longitudinal lineations developed by high-magnitude flood flows similar to erosional lineations observed in the Channeled Scabland of western Washington State, USA[42]. Taken together, the presence of a linear bedrock-eroded channel with associated cataracts, deep scours and streamlined

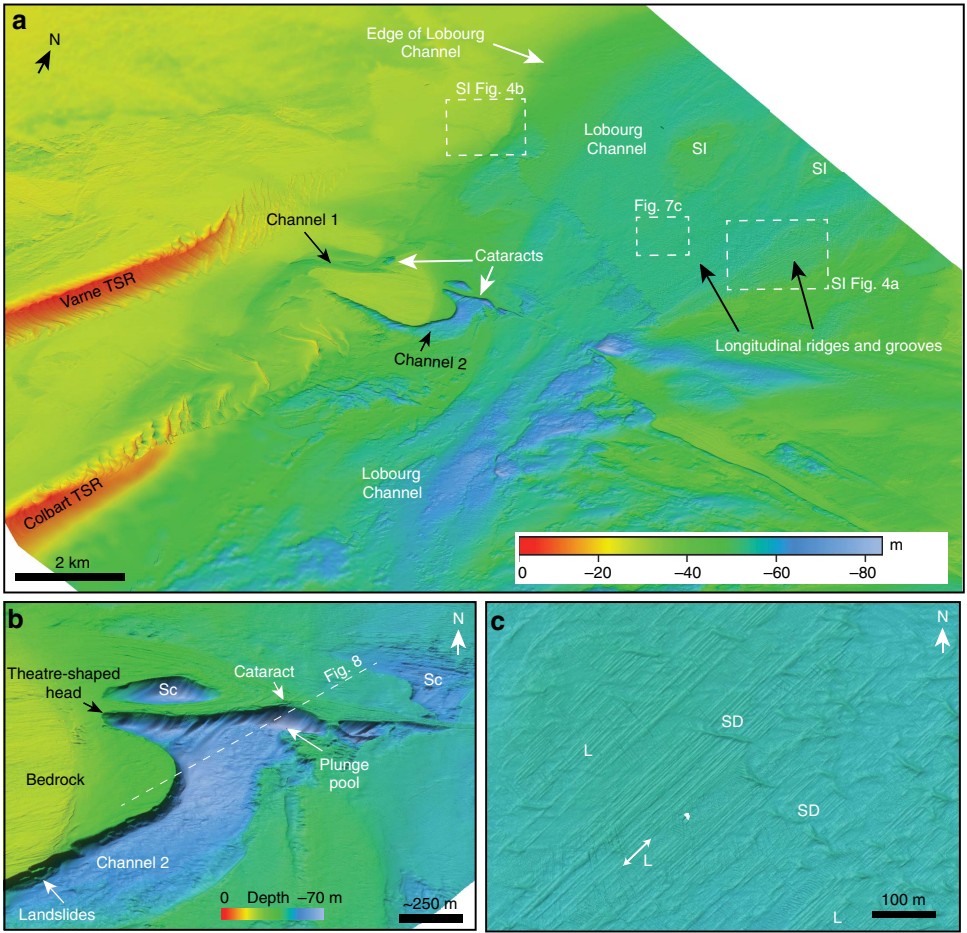

**Figure 7 | Sonar bathymetry of the central Dover Strait region.** (**a**) Coloured and shaded relief multibeam bathymetry map of the Lobourg Channel. SI, streamlined island; TSR, tidal sand ridge. Water depth is indicated by colour bar. Location of image is shown in Fig. 1. (**b**) Three-dimensional perspective view of cataract at head of Channel 2 looking N. Sc, prominent scours in bedrock. Vertical exaggeration is ∼2. Water depth is indicated by colour bar. Dashed line indicates line of seismic profile in Fig. 8. (**c**) Map view of floor of Lobourg Channel showing prominent longitudinal lineations (ridges and grooves) (L). Orientation of lineations is indicated by double-headed arrow. SD, sand dunes. Location of image is indicated on **a**. Water depth is same as in colour bar in **a**.

islands indicate that erosion of the Lobourg Channel was achieved by high-magnitude flood flows through the Strait.

## Discussion

Our marine geophysical data provide new insights into the late Quaternary landscape evolution of the Dover Strait with implications for understanding its history of breaching. In particular, the new data show that at least two significant erosional episodes shaped the opening of the Strait. While it is widely held that initial opening of the Strait was a consequence of spillover of a proglacial lake that existed in the southern North Sea basin during MIS12 (refs 2–7), direct evidence for this spillover process has up to now been lacking. The presence of large bedrock-eroded valleys with landforms associated with erosion by high-magnitude floods in the central and eastern English Channel has been proposed as evidence of catastrophic lake spillover[5,7]. Our detailed mapping of the geometry and distribution of the Fosses Dangeard depressions show that the Fosses represent plunge pool depressions formed at the base of a chalk escarpment. This provides compelling evidence of the presence of waterfalls from a proglacial lake dammed behind the Weald–Artois barrier. Spillover of lake water was clearly a key geomorphic process in the evolution of the Strait. The presence of multiple sets of depressions, some with an elongate geometry perpendicular to the strike of Cretaceous strata, suggests that erosion of the depressions may have been associated with retreat of the escarpment. A question that arises is the relationship of the Fosses Dangeard plunge pools to opening of the Strait. Did waterfall recession during lake overspill lead to eventual breaching of the escarpment and creation of the Strait? In this model, the plunge pools appear to be the only remnants of the breached dam. Given that the Fosses Dangeard depressions are distributed across much of the width of the Strait (Fig. 6), together with their scale and depth of incision, it seems likely that there were several lake spillpoints across the rock ridge characterized by significant water overspill. The anabranching bedrock-incised valley network in the central and eastern English Channel may partly be a consequence of discharges resulting from breach of the rock ridge, but our data show that this interpretation is complicated by the effects of geomorphic overprinting by later events that eroded the Lobourg Channel. The presence of a prominent valley network some 20 km southwest of the Fosse, and eroded into bedrock platform P (Fig. 2), may provide evidence of high-magnitude flows associated with breaching prior to Lobourg Channel erosion[7]. This ∼40 km long network heads on the platform and shows no connection to onshore drainage indicating that it was initiated on the platform

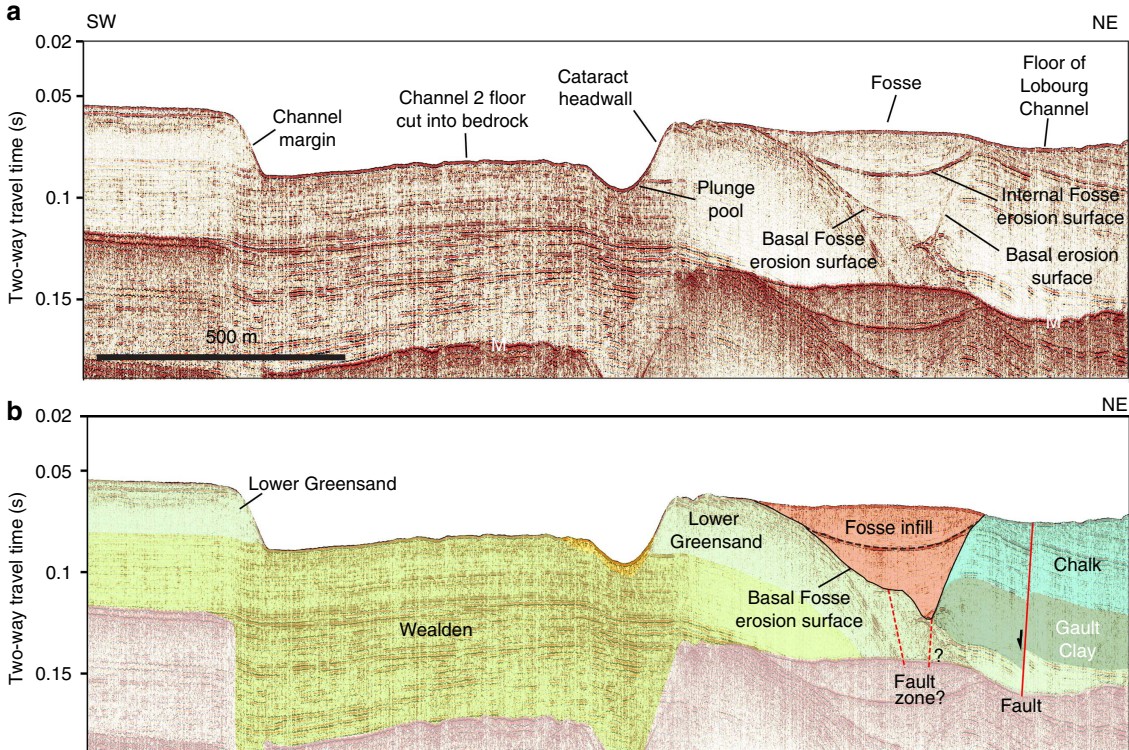

**Figure 8 | Seismic profile across inner Channel 2 and cataract headwall.** (**a**) Seismic section and (**b**) geological interpretation. A prominent scour is present at the base of the headwall that we interpret as a plunge pool. Note that the floor of the Lobourg Channel upstream of cataract cross-cuts a sediment-infilled Fosses Dangeard depression eroded into Cretaceous bedrock. Vertical exaggeration ∼4. M, seabed multiple. Location of seismic profile is indicated in Fig. 7b and Supplementary Fig. 2.

itself (Supplementary Fig. 6). In summary, we suggest that progressive recession of the chalk escarpment during lake overspill leading to eventual breaching of the dam provides a holistic model to explain initial opening of the Strait.

The combination of seismic reflection data and high-resolution bathymetry data reveals that a second event (or series of events) was needed to fully open the Dover Strait. The Lobourg Channel cross-cuts several of the Fosses Dangeard infills (for example, Fig. 8), which clearly indicates that it formed as the result of younger erosional episodes. This distinction of two stages in the opening of the Strait is further substantiated by the fact that several sets of Fosse depressions lie outside the trace of the Lobourg Channel thus discounting any genetic link between the formation of these features (Fig. 3b). The Lobourg Channel is characterized by geomorphic features indicative of erosion by catastrophic flood flows. The scale of the Lobourg Channel and presence of these characteristic landforms is suggestive that major episodes of flood flow were required to carve it.

Prior to formation of the Dover Strait, a prominent south-west-facing chalk escarpment extended across the Strait. The Channel region likely comprised a low-relief landscape during sea-level lowstands with a south-west flowing pre-breach Channel river fed primarily by river discharges from the Somme, Seine and palaeo-Solent rivers. Drainage from the escarpment to this river is likely to have been relatively small. We envisage the following sequence of events in the opening of the Strait itself: (1) spillover of pro-glacial lake water from multiple spillpoints along the chalk ridge led to excavation of several sets of plunge pool depressions, (2) retreat of the escarpment caused concomitant upstream propaga-tion of Fosses erosion, (3) erosion of the ridge at multiple spillpoints causes eventual breaching of the Strait by rock dam failure with release of dammed lake water into the eastern English Channel, (4) multiple episodes of infill and scour of the Fosses

during cycles of regression and transgression of the English Channel shelf, (5) incision of the Lobourg Channel and associated large valley network downstream by passage of a flood derived from the northeast in the North Sea basin. Erosion of the Lobourg Channel likely records the final opening of the Strait. The transition of the Lobourg Channel downstream into an anabranching network of bedrock valleys in the English Channel is suggestive of complex flood routing pathways southwest of the Dover Strait. We note that while events associated with lake overspill largely shaped the landscape of the Dover Strait, fluvial activity during lowstands with flow of a major river, the Channel River, through the Strait and marine erosion during high sea-level stands played a role in partially shaping the physiography.

While our results provide important new constraints on the relative timing of events that shaped the landscape evolution of the Dover Strait, our understanding is hampered by a lack of an accurate chronology for the sequence of events. Initial Pleistocene breaching of the Strait has been proposed to be an MIS 12 event (∼450 ka and conventionally equated to the Elsterian–Anglian Stage glaciation). The extension of the Lobourg Channel tens of kilometres northeastwards from the Strait into the present-day southern North Sea basin (Fig. 1) suggests that the flows that carved this channel were likely derived from drainage of ice-marginal lakes in the central-southern North Sea basin[6,11,19,47], or from flood flows derived from proglacial lakes further to the east in Germany[48]. The timing of these events is not directly constrained, although a MIS 6 (∼160 ka within the Saalian/ Wollstonian Stage glaciation) is plausible. Marine molluscan fauna indicate that there was sporadic interconnection between the North Sea and English Channel during some of the high sea-level stands between MIS12 and MIS 6 (refs 11,49) suggesting partial breaching of the Strait had been accomplished during this time. By MIS 5e, marine mollusc assemblages from coastal deposits indicate full

connection of the Channel with the North Sea during the interglacial highstand[26]. The two-stage evolution of the opening of the Dover Strait we propose is compatible with prior interpretation of at least two episodes of flooding from the central English Channel[5,7], and sediment records from the Celtic Sea deep sea fans[20,21]. Sediment delivery to these fans was pulsed with the greatest supply of sediment from the Channel routing system occurring during MIS stages 12 and 6 (refs 20,21). Nevertheless, the lack of *in situ* derived chronologies from the Dover Strait (and from the English Channel more broadly) represents a hindrance to understanding its opening.

The Fosses Dangeard sediment infills are an outstanding target for future drilling in order to precisely constrain the chronology of events shaping the breaching history of the Strait, and its palaeogeographic consequences. Such a chronological framework is necessary to better understand the timing of when Britain first became isolated from mainland Europe during interglacial high sea-level phases. This has profound significance to understanding the ability and timing of biota, including humans, to colonize the British Isles[11,29]. Furthermore, opening of the Strait caused large-scale re-routing of NW European drainage and meltwater to the North Atlantic via the English Channel[20,21]. The re-routing of meltwater from the British-Scandinavian Ice Sheet and its injection into the North Atlantic has implications for inter-hemispheric climate variability[22,23].

## Methods

**Marine geophysical data.** Our study uses marine geophysical data sets collected in the Dover Straits between 2002 and 2012. The locations of the individual survey areas are shown in Supplementary Fig. 2.

**Seismic reflection data.** The sub-bottom data were collected during four separate field campaigns onboard RV Sepia II and RV Belgica. Both vessels were equipped with dGPS, giving horizontal location accuracy better than 10 m.

**Single-channel seismic reflection data.** During the 2002 Seismic survey (Lille-RCMG) cruise (Dangeard 1 cruise), ∼200 km of seismic lines were acquired using a 300J Centipede sparker system triggered every 1 s, and a single-channel surface streamer consisting of 10 hydrophones. Both seismic and positioning data were logged with Delph iXBlue software. In total 20 parallel lines were collected with a ∼10 km length and ∼200 m spacing, together with one cross-profile. During the RV Belgica cruises 2010/09 and 2012/03 (2010–2012 ROB-RCMG seismic surveys) ∼315 km of single-channel high-resolution seismic-reflection profiles were collected with a 500J Centipede sparker system triggered every 0.75 s, and a single-channel surface streamer of 10 hydrophones. The data were recorded through a Delph IXSEA seismic acquisition software system. Sampling frequency was set at 10 kHz and record length at 400 ms TWT. The two surveys consisted of 48 seismic-reflection profiles spaced 1.5 km apart acquired to the southeast and northwest of the 2002 seismic survey. The single channel seismic data were processed together using the open-source package Seismic Unix (SU)[50] in conjunction with a python interface to SU (supy) developed at the Royal Observatory of Belgium. Processing consisted of: (1) conversion from SEG-Y to SU format, (2) swell filter, (3) navigation correction, (4) predictive error filtering, (5) band-pass filtering and time-varying gain, (6) muting, (7) tide correction and (8) conversion from SU to SEG-Y format. The vertical resolution of the single-channel data is 1–3 m.

**Multi-channel seismic-reflection data.** The multichannel seismic survey (2012 Seismic survey) was acquired during RV Belgica cruise 2012/25 and was carried out by means of a 24-channel streamer composed of 10 hydrophones per channel (channel spacing: 10.5 m), which was provided by the Royal Netherlands Institute for Sea Research (NIOZ). A GI-gun was used as source, fired at ∼100 bars of pressure every 7 s. Lines used either a 1 ms sample rate per 5 s record length or 0.5 ms sample rate per 2 s record length. In total, 12 profiles were collected sub-parallel to the English and French coasts with lengths ranging between 13 and 20 km and spaced between 2 and 5 km to one another. Four control cross-profiles were also acquired. The processing of the multichannel seismic-reflection data set was performed at the Renard Centre of Marine Geology (RCMG) of Ghent University using the processing software RadExPro (www.radexpro.com) by applying the following processing flow: (1) trace editing, (2) swell filtering, (3) spherical divergence corrections, (4) trace balancing, (5) amplitude decay, (6) band pass filter (30–325 Hz), (7) mild F–K filter on all offsets, (8) harsh F–K filter on far offset traces, (9) 2:1 channel interpolation, (10) pre-stack multiple

removal in the channel domain, (11) velocity analysis every 50 CMP, (12) NMO correction, (13) stack, (14) post-stack multiple removal, (15) post-stack random noise suppression and (16) post-stack time varying amplitude scaling.

**Seismic data analysis.** The processed seismic data were imported into the interpretation software Kingdom Suite (TKS) and Opendtect (http://www.opendtect.org). The Fosse Dangeard basal erosional surface and internal erosional surfaces (when possible) were correlated from one profile to another by using the three-dimensional software Opendtect. Isopach maps were built by subtracting the basal erosional surface of the various infilled depressions from the seafloor and assuming mean seismic velocity through the Fosse infill of 1,800 m s$^{-1}$. For this calculation, only the infill of the Fosse Dangeard depressions was considered. Major tidal sand ridges (for example, The Varne) were removed by subtracting the seafloor from the erosion surface at the base of the tidal sand ridges. Isopach maps are discontinuous through some of the major depressions (Fig. 3). This is due to the presence of the first seismic multiples at depths of ∼100 m, which obscures the continuation of some of the deepest depressions at depth.

**Bathymetry data.** The bathymetric data used for the study is mainly based on multi-beam echo sounding, with historical single-beam echo soundings used to complete the Dover Strait grid. All grids used a UTM Zone 31 Coordinate System.

**Multi-beam echo sounding.** The 2006/2007 MCA survey data set was collected under contract to the Maritime and Coastguard Agency (MCA) as part of their 'Civil Hydrographic Program' to IHO order 1 standards. The data were collected in 2006 and 2007 by a total of four vessels using a variety of sonar systems. The majority of the survey was collected from MV Victor Hensen with a Kongsberg EM710. This equipment operates with frequencies ranging from 60 to 100 kHz. Later infill lines were collected from MV Meridien (Reson SeaBat 8101ER and the SeaBat 7125 with operating frequencies of 240 and 400 kHz, respectively); MV Geniusbank (Kongsberg EM3002D with operating frequency of 300 kHz) and MV Jetstream (also equipped with a Kongsberg EM3002D). The merged data set was provided to the Imperial College of London pre-processed and tide-corrected in a GSF file format. The data were gridded in IVS 3D DMagic with a 1.5 m cell size. The 2010 ROB–RCMG survey data set was collected in April of 2010 on board of the RV Belgica with a hull-mounted Kongsberg EM3002 multi-beam echo-sounder operating at 300 kHz. Sound absorption coefficient was calculated in the survey area from the temperature, salinity and pH of the water given by an ODAS-II system. Based on the estimated sound absorption coefficient and a sound-velocity profile that was taken with the SV-plus probe on 15.04.2010 at 51.025° N, 1.51° E, an average sound velocity of 1,483 m s$^{-1}$ was selected for the water layer. These data were subjected to statistical spike removal, manual cleaning of artefacts and tide correction. The tide correction was performed by applying information provided by the Hydrographic and Oceanographic Centre of the French Marine (SHOM) on the water elevation at the different times and locations surveyed. No special filtering was necessary due to the good quality of the raw data. The corrected bathymetry set was gridded at 5 m cell size at the Renard Centre of Marine Geology using the Sonar Scope software.

**Single-beam echo sounding.** Historical single-beam echo soundings of the UK sector of the English Channel collected by the Maritime and Coastguard Agency (MCA) between 1988 and 2004 and archived at the UK Hydrographic Office (UKHO) were provided to Imperial College London. The soundings were gridded using the software QPS Fledermaus DMagic by G. Potter in November 2009 with a 30 m cell size. Historical single-beam echo soundings of the French sector of the English Channel (La Manche) collected by the Service Hydrographique et Océa-nographique de la Marine (SHOM) between 1973 and 2000 were provided to the University of Lille 1 under contract 205/2011. The data were gridded at the University Lille 1, France, using QPS Fledermaus DMagic in July 2012. The density of the soundings in the data set was highly variable so two grids were produced, one with a 80 m cell size and a second with a 40 m cell size.

**Bathymetry merge.** The final bathymetric map was produced using QPS Fledermaus DMagic and Global Mapper 14. Individual Arc Grid files (.asc) from the previously gridded .sd files were used for the 'scene file merge'. The Arc Grid files were loaded into Global Mapper in a single workspace and overlapped in order to display the data set at the highest resolution in each area. The workspace comprising all the different grids was then saved into a single Arc Grid file respecting the different cell size of each of the grids composing the map.

**Data availability.** The data that support the findings of this study are available from the corresponding author upon reasonable request. Details on Dangeard-1-cruise data that support the findings of this study are available from Sismer through the following link http://dx.doi.org/10.17600/2440010. SHOM bathymetric data at 20 m resolution is available at http://diffusion.shom.fr/produits/bathymetrie/mnt-cotier-pas-de-calais.html UKHO data is available as a ∼400 m DEM at http://portal.emodnet-bathymetry.eu/?menu=0310000_000000.

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

## Acknowledgements

The UK bathymetry surveys were gathered under the UK Civil Hydrography Programme managed by the Maritime and Coastguard Agency (MCA) and the United Kingdom Hydrography Office (UKHO). We thank A. Talbot (UKHO) for assistance. Shiptime on RV Belgica for bathymetric and seismic surveys was provided by the Belgian Science Policy Office and the Management Unit of the North Sea Mathematical Models (MUMM). We thank the Captains and Crews of CNRS/INSU RV Sepia II and RV Belgica for their assistance in data acquisition. We acknowledge the support of FWO-Flanders and the Special Research Fund of Ghent University. Service Hydrographique et Océanographique de la Marine (SHOM) provided single-beam data for French waters (Contract No 205/2011). We thank Eurotunnel for permission to use geophysical data from the 1986 and 1988 Channel Tunnel seismic studies. S.G. was partially supported by a Royal Society-Leverhulme Trust Senior Research Fellowship. T. Henstock, W. Versteeg, H. Jomard and K. Verbeeck are thanked for support. We thank K. Cohen, F. Busschers, B. Coakley and J. Scheingross for discussions, and P. Gibbard and D. Hodgson for reviews.

## Author contributions

S.G. and J.S.C. conceived, designed and coordinated study. J.S.C. and F.O. processed bathymetry data. G.P. compiled and processed UK singlebeam bathymetry data. D.G.-M.

processed ROB-RCMG bathymetry data. A.T. and B.V.V.L. coordinated RV Sepia II seismic survey. T.C. and M.D.B. designed and coordinated RV Belgica seismic surveys. K.V. and D.G.-M. coordinated and conducted data acquisition and processing. J.C.R.A. provided input on Channel Tunnel surveys. S.G., D.G.-M., F.O., J.S.C., A.T. and M.D.B. analysed and interpreted geophysical data sets. All authors discussed the results. S.G. and J.S.C. wrote the paper with important contributions from D.G.-M.

## Additional information

**Competing interests:** The authors declare no competing financial interests.

