## [Peer Review File · Nature Communications]

Reviewers' Comments:

Reviewer #1 (Remarks to the Author)

Gupta, S. et al.. Making Britain: two-stage catastrophic opening of the Dover Strait.

This article presents an examination of the nature of the floor of the Dover Strait (Pas de Calais) between Britain and France. The region is significant since it represents the link between the southern North Sea basin and the English Channel (la Manche), the formation of which has provoked the attention of workers for two centuries. However, it was not until 30 years ago when Professor Smith offered a credible mechanism for the formation of the palaeochannel system in the eastern English Channel as well as the breach across what had previously been the Weald-Artois anticlinal ridge. He proposed that the ridge could have been breached by a catastrophic flood, although he failed to offer a plausible explanation of the mechanism that initiated the flood event. Later workers have demonstrated that the flood arose from overtopping of the Weald-Artois bedrock ridge by overspilling of a massive glacial lake in the southern North Sea basin, formed in the Middle Pleistocene when glacial ice advanced from Northern Britain and Scandinavia, blocking the basin exit to the North Atlantic. Initially this was seen as a single event, but later investigations in the southern North Sea region confirmed that a second major lake formed during a younger glaciation. The recognition that the overflow from these lake entered the Channel River (Fleuve Manche) by two of the authors of the current article, led to confirmation that the floods were indeed catastrophic in nature, leaving a legacy of characteristic landforms on the Channel floor. Despite this confirmation of the broad mechanism for the catastrophic formation of the breach, detail of the nature of the erosional breakthrough the Weald-Artois ridge has remained elusive, because, as the authors rightly state, the breach is a zone of substantial erosion, making evidence difficult to recover. However, the present article offers some startling and exciting new insights into the detailed formation of the breach.

By collecting and interpreting new high-resolution sonar-derived bathymetry and seismic reflection data from the Strait floor based on sub-bottom records, the authors present striking evidence of a series of erosional landforms that cast new light on the process of the breakthrough of the Weald-Artois ridge. Whilst it is a major achievement even to collect these high-resolution records in the busiest sea-lanes in the World - no mean undertaking in itself! - the authors demonstrate an assemblage of isolated, sediment-infilled depressions that are deeply incised into bedrock. These substantial features appear to be aligned along the Chalk exposure and are interpreted as massive potholes, thought to have been formed by cascading, highly turbulent water.

As stated earlier, previous workers have identified two major breaching and 'megaflooding' episodes, relating to different Middle Pleistocene glacial events (the first in the Elsterian/Anglian Stage, and the second in the Saalian/Wolstonian - cf. comment below). The analytical results presented by the authors demonstrate that a second event certainly occurred and indeed it was required to complete the opening of the Strait. The valley that currently occurs on the floor of the Strait today, the so-called Lobourg Channel, originated as a consequence of this second major flooding event, the floor of the channel including erosional features characteristic of catastrophic flooding process. Interestingly the Lobourg Channel appears unrelated to the giant pothole structures, indicating that its formation post-dates the latter forms. As the authors conclude, later modification of the Strait by flood erosion of the Lobourg Channel may have resulted from overflow of younger ice-marginal lake systems in the North Sea basin or beyond (cf. comment below).

It is beyond question that these events had profound impacts on the drainage of the western

European region through the Middle to Late Pleistocene and have resulted in Britain's isolation (in all senses!) from the Continent during interglacial high sea-level phases, like today. These impacts extend far beyond the physical aspects of isolation, but also profoundly influence the ability of the biota, including human, to colonise the British islands.

I have no significant criticisms of the science in this article, and must stress that this is an interesting and impressive investigation that reinforces the previous interpretations by the first two authors of this article. However, it unquestionably offers important and new evidence that greatly extends our knowledge of the detailed character of the ridge-breaching process, as well as its relative timing. As the authors say, there is clearly much more to do, but this requires horrendously expensive, and highly risky drilling to recover the sediments filling these deep depressions identified here. Clearly, this coring might never happen because of the sheer difficulty and danger of coring in the shipping lanes. However, the present article goes a very long way to offering us valuable information that is both new and novel.

This is an important paper, which is well presented and the results are scientifically balanced and clearly interpreted. The illustrations are excellent and appropriate. The topic is highly suitable for publication in *Nature*, and the discussion is undoubtedly of general interest. The paper requires only very minor editorial modification before it being accepted for publication.

Here I offer a few comments for the authors to consider or correct as appropriate.

p.3 In the initial discussion of previous work, the authors refer to my article in 1995 on the formation of the Dover Straits, but I notice that they have not cited my earlier article on the breaching published in 1988:

Gibbard, P.L. 1988. The history of the great northwest European rivers during the last three million years. *Philosophical Transactions of the Royal Society of London* B318, 559-602.

I realise they are restricted to a limited number of references, but this article is relevant to their discussion.

p. 6 l. 135 and p. 7 l.138. What is 'theatre-shaped'? I do not understand.

p.7 l. 188-189 Here the text reads "Initial Pleistocene breaching of the Strait is proposed to be an Elsterian/Anglian event. Drainage of younger, Saalian-age or last glacial ice-marginal lakes in the southern North Sea basin". The stratigraphical terminology should be tightened here a little. The terms Elsterian/Anglian are (chronostratigraphical) stages, as is Saalian, but this should, for consistency say Saalian/Wolstonian (the latter being the British equivalent to Saalian) and the "last glacial" should read Weichselian/Devensian.

p.8 l. 200 etc. "Finally, whilst a flood erosion model can readily explain the extensive ridge groove features on the floor of the Lobourg Channel, their linearity and striking resemblance to streamlined glacial lineations observed in digital topographic datasets from deglaciated regions also opens the possibility that a younger episode of palaeo-ice-stream flow was superposed on the flood-eroded landscape in the Dover Strait. Multi-beam datasets to test this are limited in the Strait, but if correct, and given that the lineations are found on both the floor and flank of the Lobourg Channel (Extended Data Fig. 6), this requires a Saalian or younger surge of a tongue of the British-Scandinavian Ice Sheet ~200 km further south than previously considered, with implications for its size and transient behaviour".

I found myself taking a deep breath at this statement! The "surge of a tongue of the British-Scandinavian Ice Sheet ~200 km further south than previously considered" reminded me of the previous, discredited ideas invoked by Kellaway and others in 1975 of glaciation of the Channel (also referred to in the article, l. 55). Whilst these lineations are certainly striking, there are other

potential explanations which should be considered. One possibility are tidal sand ridges which are widespread on the floor of both the North Sea and Channel. Another possible explanation is iceberg keel drag or plough marks. These are certainly commonly found in glaciated marine settings throughout the world. Could these marks result in the same way? It is important not to set the 'hare running' on glaciation of the Straits area based on such tenuous evidence that has not been fully investigated. I find the concept of glaciation of the Dover Strait implausible, particularly in the absence of any other supporting evidence of glaciation so far south in both the North Sea and Channel regions, as far as one can see. of course, I could be wrong!

It is perhaps also worth stressing that the Strait is a polygenetic feature in the sense that it has developed, not only through glacial overspill of two, and possibly a third glacial lake dammed in the North Sea basin (the third being during the Weichselian/Devensian Stage, cf. Woldstedt 1960), but also the evolution is related to substantial fluvial activity during lowstand phases, when the strait was occupied by the Channel River. The latter undoubtedly persisted for long periods, i.e. thousands of years during the non-glacial intervals of the cold periods (i.e. c. 200 ka in the Saalian/Wolstonian and c. 100 ka in the Weichselian/Devensian stages). By contrast, during high sea-level stands in the interglacials the tidal and coastal erosion of the Strait has also been significant and severe, as can be seen from the actively eroded, cliffed coastlines on both sides of the Strait today.

In conclusion, this is an interesting, important, stimulating contribution that is a welcome addition to our knowledge. It is certainly acceptable for publication after minor modification.

P.L.Gibbard
Cambridge
30.7.2016

Reviewer #3 (Remarks to the Author)

This is a most interesting and topical paper. it is well written and makes a significant contribution to the subject. I recommend publication with minor revision.

All my specific comments are on the attached pdf and most relate to the Quaternary context and ages. I am of the opinion that the geomorphology is good and well supported by the geophysical studies.

My general comments are:

- 1) Determination of the age of the formation of Stage 2 (Loberg Channel. Apart from determining its separation from the initial Stage 1, its age is open-ended. The separation is important and justifies the publication of this paper, but a more authoritative discussion could arise from the careful discussion of Toucanne's work.
- 2) Consideration of the glacial lineaments. This is really a throw-away component of the paper. It needs a little more authority as to why glaciation is not likely to be the cause. Although these features are parallel, like glacier lineaments (lineations, they do not have the smooth, streamlined edges of glacial features, and look more like water shaped bedforms.
- 3) A map is needed to show the context of the site in terms of North European (including BI) ice sheet limits.

I have been through the captions of figures and Supplementary Text and have nothing to add

Review of Making Britain: Two-stage catastrophic opening of the Dover Strait by Gupta et al. [NCOMMS-16-16909-T]

Summary:

The original (physiographic) Brexit – I can see the tweets already – although the two-stage erosional opening interpreted here is more efficient than our future political detachment. Previous interpretations on the processes and timing of the opening of the Dover Straits have largely relied on bathymetric data, and therefore the interpretation of submerged landforms, to discuss the interplay of process responsible for the opening of the English Channel/La Manche. Therefore, it is great to see some new high resolution geophysical data presented from the distinctive Fosse-Danegard area. The distinctive erosional landforms in this area are close to the inferred spillpoint of a proglacial lake. Catastrophic drainage of this lake is the presumed source for megaflood(s) that have been interpreted to be responsible for the erosion of a chalk escarpment that previously attached Britain and France. Therefore a better constraint for the controls on the formation, and filling, of these landforms are critical for understanding the timing and environmental impact of connecting southern and northern water masses.

In many ways, the authors do not move far from the hypothesis postulated by Smith (1985) on the processes involved in the formation of the English Channel. These data do not have any lithological calibration. However, the major contribution is the proposal of a refined sequence of events that led to the opening of the Dover Straits. A key new insight is the two-stage phase of erosion, linked to catastrophic flooding, and the formation of a younger Lobourg Channel.

I found the initiation mechanisms invoked and age relations of the Lobourg Channel to be convincing. However, I think that, despite the constraints of the short format, they need to make some more room to present more details on the actual processes during formation of the Fosses, the actual evidence for the breach (the plunge pool as evidence need much more discussion if they are evidence of waterfalls but not a breach), and more on the bedrock character and pre-existing tectonic features. The room to expand can be most easily found by chopping out the provocative but speculative section on the linear ridges that ends the paper. In the absence of any associated evidence of ice this far south in the Saalian, I found this section to undermine the primary aims of the paper.

Is this suitable for publication in Nature Comms.? Yes, I think so – the data has been long known about, so it is important to see it published in a leading journal. These data are well presented, and will be an impactful contribution to the journal. However, I recommend a major revision where the authors consider and present the landscape evolution more clearly, what the new data shows that is new, discuss the formation of plunge pools (non-exceptional formation) and evidence for pre-Lobourg breach, and are clearer on the geological context and process evolution to support their assertion of a two-stage megaflood.

Issues:

1. Comparison to Smith (and others) work: I was struck by how similar the Fosse distribution here is to that presented by Smith (1985) (see simple overlays below). That is good, and shows what a great job Smith did with presumably less good/comprehensive data. However, this does raise the question of what does the new geophysical data show. Therefore, the

authors need to be much clearer on this. Is it a refined or more accurate distribution of Fosse? Is there better constraints on their shape and orientation? The age relations with the Lobourg channel are compelling, but need to be clearer about the new insights in Fosse formation. Seismic facies on the infill of the Fosse is largely new, but this is not the focus here

Smith (1985) distribution of deeps/fosses (crudely) place on distribution presented here

2. Tectonic control: The authors casually dismiss a tectonic control (line 100-102) without due discussion. Indeed they state that there is horizontal strata despite they own data presented showing that this is simply not the case. For example, Fig 2a does not have horizontally stratified strata, and hard to reconstruct to the pre-Fosse development without the presence of a normal fault coincident with the upstream part of the Fosse. Currently, data and arguments from earlier workers, such as Colbeaux et al. (1980) are not considered. So, while the evidence for active tectonism during the opening of the straits is weak, the presence, orientation, and type of tectonic structures associated with lithological boundaries and sites of erosion should be considered as a control on Fosse development and distribution.
3. Processes: Drilling is not a term I have heard/seen used in relation to erosional processes associated with waterfalls. If it is, please make reference, although I failed to find any paper on a quick literature search. In bedrock fluvial systems terms such as plucking, abrasion, and cavitation are commonly used. Indeed, there is very little actually process explanation for the development of these plunge pools, or characterisation of the material being eroded.
4. Slow but deep erosion: This leads me onto a very interesting paper by Anton et al. (2015) Nat. Comms., on similarly deep, and still forming, plunge pools that formed from moderate ('unexceptional') floods. They demonstrate that in a man-made dam in Spain a large amount of erosion, and plunge pool formation, have occurred without major floods. This is an alternative mechanism to explore in relation to the formation of the Fosse Dangeard, although I did not find it clear whether the authors though the plunge pool formation was part of the landscape response to the inferred megaflood, or a phase prior to breaching of the rock dam.
5. Alternative model: The plunge pools themselves do not support breaching of the rock dam. They are compelling evidence of the presence of waterfalls from a proglacial lake, which was partially confined by a topographic high ridge formed by the chalk. However, the authors seem to make a jump from these to a breach and megaflood (although other places they are careful to make these distinct. It is not clear, but I think the authors feel that the scale of plunge pools need to be related to a flood event, but elsewhere they intimate that their formation was a phase prior to breach. If there was a breach, how did these get preserved? Does their preservation suggest the breach was to the south? The authors can do more to paint an image of the landscape pre- (and post-)breach. There was an escarpment, and for

the plunge pools to form, it was a lowstand, and the waterfalls were likely ephemeral (as in a man-made and controlled dam), there may be streams feeding the lake at low levels down the NE-facing slope. Gibbard does a good job of this. Is there evidence in the fosse that there was escarpment retreat? In other words, is the formation, and possibly filling of D, F C earlier than E and A. This could be supported by their different planform geometries. But why invoked a breach at this stage? Could there have been only one breach, which occurred in a later phase, possibly to coincide with the formation of the Lobourg Channel. The phase of plunge pool development may have lasted sometime - and in response to different periodicities of ice melt and lake level. The infill of the Fosse was also possibly over shorter durations - how many episodes, of scouring and filling, and how many glacial-interglacial cycles since the Elsterian.... suggests shorter duration of infill.

6. Fosse mapped surface: I didn't note in the discussion or methods whether there was evidence for the Fosse being on a single composite erosion surface, or within the constraints of the data, could their formation be multiple ages?
7. Internal surfaces: hard to see from the data presented the orientation of the scoop. Are these scour forms with steep headwall and shallowing and flaring downdip? These are interesting, as difficult to envisage flows undergoing a hydraulic jump within the confines of a plunge pool.
8. Fosse morphology: 'C' looks more elongate NE-SW, whereas A, B, E are more elongate about NW-SE. Hard to see from data on D and F, yet in the text you describe them as similar in orientation a shape. Feels like there is more to get at here with the distribution of different orientations and depths with respect to underlying bedrock and tectonic features.
9. Timing for non-experts: For the non-experts in the (somewhat opaque) world of Quaternary chronostratigraphy that might be reading, worth putting in some approximate years BP (and include Anglian and Wolstonian) to help accessibility of the text.
10. Geology for non-experts: What is the bedrock? This is labelled on the figures, and most UK geologists will know what underlies the chalk, which is strangely the only rock mentioned. You need to be more explicit in the text that the plunge pools, and other erosional bedforms, have been cut into the Gault Clay, and sometimes the Lower Greensand where deep enough. This is important for accessibility, but more so to characterise the bedrock to help discussion of the process or formation and location. The Gault Clay is less resistant compared to the Lower Greensand. Other authors have shown the important controls the bedrock geology has, but only the chalk is mentioned here – needs more clarity.
11. Other processes: The authors used terms like 'the megaflood landscape', which is misleading. Many authors have shown/argued, the landscape/seascape is a palimpsest, and is a physiographic record of a wide range of erosional and depositional processes. Even if there were one or several large flood events, there has been significant modification by other processes, fluvial, wave, tide.
12. Linear ridges: The linear ridges do look very like the basal surface of submarine landslides - so the flood event origin, with a high concentration flow, is possible. Without any other evidence for glacial activity in this section this is highly speculative. Therefore, I'm not keen on this final paragraph - provocative is fine, but too speculative. A stronger paper if they strengthen their assertions on the number and mechanisms of floods through more discussion earlier.

RESPONSE TO REVIEWERS COMMENTS

We thank Prof. Gibbard for his detailed comments on the manuscript and his recognition of the significance of our new results.

Response to reviewer P. Gibbard

Here I offer a few comments for the authors to consider or correct as appropriate.

p.3 In the initial discussion of previous work, the authors refer to my article in 1995 on the formation of the Dover Straits, but I notice that they have not cited my earlier article on the breaching published in 1988:

Gibbard, P.L. 1988. The history of the great northwest European rivers during the last three million years. *Philosophical Transactions of the Royal Society of London* B318, 559-602.

I realise they are restricted to a limited number of references, but this article is relevant to their discussion.

Added reference

p. 6 l. 135 and p. 7 l.138. What is 'theatre-shaped'? I do not understand.

Changed to amphitheatre-shaped

p.7 l. 188-189 Here the text reads "Initial Pleistocene breaching of the Strait is proposed to be an Elsterian/Anglian event. Drainage of younger, Saalian-age or last glacial ice-marginal lakes in the southern North Sea basin". The stratigraphical terminology should be tightened here a little. The terms Elsterian/Anglian are (chronostratigraphical) stages, as is Saalian, but this should, for consistency say Saalian/Wolstonian (the latter being the British equivalent to Saalian) and the "last glacial" should read Weichselian/Devensian.

Terminology has been improved in text.

Glacial lineations

p.8 l. 200 etc. "Finally, whilst a flood erosion model can readily explain the extensive ridge groove features on the floor of the Lobourg Channel, their linearity and striking resemblance to streamlined glacial lineations observed in digital topographic datasets from deglaciated regions also opens the possibility that a younger episode of palaeo-ice-stream flow was superposed on the flood-eroded landscape in the Dover Strait. Multi-beam datasets to test this are limited in the Strait, but if correct, and given that the lineations are found on both the floor and flank of the Lobourg Channel (Extended Data Fig. 6), this requires a Saalian or younger surge of a tongue of the British-Scandinavian Ice Sheet ~200 km further south than previously considered, with implications for its size and transient behaviour".

I found myself taking a deep breath at this statement! The "surge of a tongue of the British-Scandinavian Ice Sheet ~200 km further south than previously considered" reminded me of the previous, discredited ideas invoked by Kellaway and others in 1975 of glaciation of the

Channel (also referred to in the article, l. 55). Whilst these lineations are certainly striking, there are other potential explanations which should be considered. One possibility are tidal sand ridges which are widespread on the floor of both the North Sea and Channel. Another possible explanation is iceberg keel drag or plough marks. These are certainly commonly found in glaciated marine settings throughout the world. Could these marks result in the same way? It is important not to set the 'hare running' on glaciation of the Straits area based on such tenuous evidence that has not been fully investigated. I find the concept of glaciation of the Dover Strait implausible, particularly in the absence of any other supporting evidence of glaciation so far south in both the North Sea and Channel regions, as far as one can see. of course, I could be wrong!

We have removed this discussion based on reviewer comments, though we note that the lineations do not resemble iceberg keel marks which are very different in morphology (see Newton et al. Nature Comms 2016). Nor do they resemble tidal sand ridges which we can clearly discern in the data as separate entities. Moreover, the lineations are eroded into bedrock and thus cannot be tidal sand ridges.

It is perhaps also worth stressing that the Strait is a polygenetic feature in the sense that it has developed, not only through glacial overspill of two, and possibly a third glacial lake dammed in the North Sea basin (the third being during the Weichselian/Devensian Stage, cf. Woldstedt 1960), but also the evolution is related to substantial fluvial activity during lowstand phases, when the strait was occupied by the Channel River. The latter undoubtedly persisted for long periods, i.e. thousands of years during the non-glacial intervals of the cold periods (i.e. c. 200 ka in the Saalian/Wolstonian and c. 100 ka in the Weichselian/Devensian stages). By contrast, during high sea-level stands in the interglacials the tidal and coastal erosion of the Strait has also been significant and severe, as can be seen from the actively eroded, cliffed coastlines on both sides of the Strait today.

Yes we have made this a little clearer. We have added a sentence on this to the text.

Response to reviewer 3

This is a most interesting and topical paper. It is well written and makes a significant contribution to the subject. I recommend publication with minor revision.

We thank the reviewer for these comments.

All my specific comments are on the attached pdf and most relate to the Quaternary context and ages. I am of the opinion that the geomorphology is good and well supported by the geophysical studies.

My general comments are:

1) Determination of the age of the formation of Stage 2 (Loberg Channel). Apart from determining its separation from the initial Stage 1, its age is open-ended. The separation is important and justifies the publication of this paper, but a more authoritative discussion could arise from the careful discussion of Toucanne's work.

We have added some more discussion of this but have not ventured to go too deeply as it becomes too speculative without independent chronological constraints.

2) Consideration of the glacial lineaments. This is really a throw-away component of the paper. It needs a little more authority as to why glaciation is not likely to be the cause. Although these features are parallel, like glacier lineaments (lineations), they do not have the smooth, streamlined edges of glacial features, and look more like water shaped bedforms.

We have removed this section.

3) A map is needed to show the context of the site in terms of North European (including BI) ice sheet limits.

Done – added Supplementary Figure 1

Comments on pdf

1. Title – changed to **“Two stage opening of the Dover Strait and the origin of ‘island’ Britain”**
2. Line 102 “Why are they not solution hollows in Chalk”

The Fosse depressions do not only occur in Chalk but are largely cut into Lower Greensand sandstones and Weald Clay - so solution hollows is not a reasonable model.

3. line 171: **removed mention of glacial lineations**
4. line 188-189: **improved precise definition of ages in terms of MIS and provided appropriate referencing**

Response to Hodgson review comments

We thank Prof. Hodgson for his very detailed comments. These have been most useful to us in revising the manuscript and clarifying some key points, in particular with the expanded length available in Nature Communications.

In many ways, the authors do not move far from the hypothesis postulated by Smith (1985) on the processes involved in the formation of the English Channel.

We clearly state we set out to test the hypothesis of Smith (1984). However, Smith did not present any primary data in his paper nor discuss the data sources of his figures in detail. None of the primary seismic data is presented, nor details of how the map of the Fosse generated. Smith has taken a map of the Fosse originally developed by Destombes et al. (1975) as part of the Channel Tunnel surveys. Also, in the Destombes et al. (1975) paper, none of the primary seismic data is presented nor details of where the seismic tracklines are located. It is not possible to reconstruct how the morphology of the Fosse was constructed in this original study, which were interpreted the Fosse as sub-glacial tunnel valleys.

We made much effort to compile all seismic data related to the Channel Tunnel surveys. We had no success in recovering the data used by Destombes et al. (1975) and this data is assumed lost (D. Blundell, pers. comm).

As a result, it is very difficult to scientifically assess the ideas of Smith from his 1984 paper. The paper is brilliant but speculative and was thus not broadly accepted by the community. We suspect that this is largely because the paper did not contain any primary data.

Our study is the first to collect new data since the 1970 and 1980s to tackle this problem, and as such we do consider that this is a major step forward.

These data do not have any lithological calibration. However, the major contribution is the proposal of a refined sequence of events that led to the opening of the Dover Straits. A key new insight is the two-stage phase of erosion, linked to catastrophic flooding, and the formation of a younger Lobourg Channel.

Agreed. These relationships were not at all apparent in the presentation by Smith. He considered that the origin of the Fosse Dangeard and Lobourg Channel were related.

I found the initiation mechanisms invoked and age relations of the Lobourg Channel to be convincing. However, I think that, despite the constraints of the short format, they need to make some more room to present more details:

- on the actual processes during formation of the Fosses,
- the actually evidence for the breach (the plunge pool as evidence need much more discussion - they are evidence of waterfalls but not a breach),
- and more on the bedrock character and pre-existing tectonic features.

These issues have been considered in much more detail now in the manuscript in particular because we are able to expand the length of the paper. A new section has been added on the bedrock character and pre-existing tectonic features, there is detailed discussion of the formation processes of the plunge pools and its relationship

to a breach in the Discussion.

The room to expand can be most easily found by chopping out the provocative but speculative section on the linear ridges that ends the paper. In the absence of any associated evidence of ice this far south in the Saalian, I found this section to undermine the primary aims of the paper.

Agreed. This section has been removed.

However, I recommend a major revision where the authors consider and present the landscape evolution more clearly, what their new data shows that is new, discuss the formation of plunge pools (non-exceptional formation) and evidence for pre- Lobourg breach, and are clearer on the geological context and process evolution to support their assertion of a two-stage megaflood.

We believe the new expanded manuscript covers the above suggestions.

Issues:

1. Comparison to Smith (and others) work: I was struck by how similar the Fosse distribution here is to that presented by Smith (1985) (see simple overlays below). That is good, and shows what a great job Smith did with presumably less good/comprehensive data. However, this does raise the question of what does the new geophysical data show. Therefore, the authors need to be much clearer on this. Is it a refined or more accurate distribution of Fosse? Is there better constraints on their shape and orientation? The age relations with the Lobourg channel are compelling, but need to be clearer about the new insights in Fosse formation. Seismic facies on the infill of the Fosse is largely new, but this is not the focus here

Neither Smith (1985) nor Destombes et al (1975) (from which Smith derived his map) present any primary geophysical data. This, together with the absence of location of seismic tracklines makes it difficult to know how the map of Fosse distribution was constructed and on what observational basis. Without the raw data the ideas could not be substantiated. As mentioned above we made efforts to track down original data but these appear to be lost.

Our new data provide a much more detailed and accurate map of the distribution of the Fosse and the presentation of the primary seismic data provides significant new detail that simply could not be discerned in Smith's presentation. The only figure

Smith showed in his paper relating to the Fosse is the one presented above. No seismic data or even line drawings of the Fosse cross-sections were presented making it difficult to test the veracity of the claim the Fosse represented plunge pools.

Overall the broad geometry of the Fosse is similar, however in detail we present a more refined and accurate morphology of the Fosse. The individual Fosse are more clearly identified, for example, Fosse D and B were not previously identified. Their shape and orientation is better defined. Moreover a detailed isopach map of thickness of sediment infill in the Fosse is presented. Smith showed the Fosse extending downstream into a series of infilled palaeovalleys. Our data show that this is not correct. We do not find filled palaeovalleys as indicated in his map (above) in our data. A major new aspect of our paper is the integration of the bathymetry data with the seismic data which leads to the identification of two separate events in the formation of the Strait – Smith (1985) identified only one.

2. Tectonic control: The authors casually dismiss a tectonic control (line 100-102) without due discussion. Indeed they state that there is horizontal strata despite their own data presented showing that this is simply not the case. For example, Fig 2a does not have horizontally stratified strata, and is hard to reconstruct to the pre-Fosse development without the presence of a normal fault coincident with the upstream part of the Fosse. Currently, data and arguments from earlier workers, such as Colbeaux et al. (1980) are not considered. So, while the evidence for active tectonism during the opening of the straits is weak, the presence, orientation, and type of tectonic structures associated with lithological boundaries and sites of erosion should be considered as a control on Fosse development and distribution.

A new section has been added to discuss this in the paper – lines 230-238. Colbeaux et al. (1980) says little with regard to the Strait and is not really relevant to our paper. The structure has been discussed in detail by Garcia-Moreno et al. (2015)

3. Processes: Drilling is not a term I have heard/seen used in relation to erosional processes associated with waterfalls. If it is, please make reference, although I failed to find any paper on a quick literature search. In bedrock fluvial systems terms such as plucking, abrasion, and cavitation are commonly used. Indeed, there is very little actual process explanation for the development of these plunge pools, or characterisation of the material being eroded.

‘Drilling’ is referenced in the following which we cited lower down in the text:

Lamb, M. P., Howard, A. D., Dietrich, W. E. & Perron, J. T. Formation of amphitheater-headed valleys by waterfall erosion after large-scale slumping on Hawai'i. *Geological Society of America Bulletin* 119, 805-822, doi:10.1130/b25986.1 (2007).

We have now cited this in reference to ‘drilling’. Without being speculative it is difficult to say much more than we have about process explanation for the plunge pools though we have added a little more detail. We have added text on the character of the material in the section on bedrock geology.

Slow but deep erosion: This leads me onto a very interesting paper by Anton et al. (2015) Nat. Comms., on similarly deep, and still forming, plunge pools that formed from moderate ('unexceptional') floods. They demonstrate that in a man-made dam in Spain a large amount of erosion, and plunge pool formation, have occurred without major floods. This is an alternative mechanism to explore in relation to the formation of the Fosse Dangeard, although I did not find it clear whether the authors thought the plunge pool formation was part of the landscape response to the inferred megaflood, or a phase prior to breaching of the rock dam.

We believe the plunge pool formation occurs prior to breaching but have now clarified this in the text in the Discussion. This paper (reference now added) is interesting but the features we describe are much larger in scale and there are many more of them. Multiple spillways were clearly present.

4. Alternative model: The plunge pools themselves do not support breaching of the rock dam. They are compelling evidence of the presence of waterfalls from a proglacial lake, which was partially confined by a topographic high ridge formed by the chalk. However, the authors seem to make a jump from these to a breach and megaflood (although other places they are careful to make these distinct. It is not clear, but I think the authors feel that the scale of plunge pools need to be related to a flood event, but elsewhere they intimate that their formation was a phase prior to breach. If there was a breach, how did these get preserved? Does their preservation suggest the breach was to the south? The authors can do more to paint an image of the landscape pre- (and post-) breach. There was an escarpment, and for the plunge pools to form, it was a lowstand, and the waterfalls were likely ephemeral (as in a man-made and controlled dam), there may be streams feeding the lake at low levels down the NE-facing slope. Gibbard does a good job of this.

We have now more carefully separated the formation of the plunge pools from the breach (lines 507-548). The expansion of the paper to now almost double its original length enables us to explain our preferred model more carefully. The relationship of plunge pool erosion and the subsequent breach is discussed in detail in the Discussion section now (and pulled out of the Results)

Is there evidence in the fosse that there was escarpment retreat? In other words, is the formation, and possibly filling of D, F C earlier than E and A. This could be supported by their different planform geometries. But why invoked a breach at this stage? Could there have been only one breach, which occurred in a later phase, possibly to coincide with the formation of the Lobourg Channel. The phase of plunge pool development may have lasted sometime - and in response to different periodicities of ice melt and lake level.

Escarpment retreat is now discussed in the paper. We now discuss why we invoke a breach in the Discussion.

It is difficult to explain the breach as coinciding with formation of the Lobourg Channel. The formation of the Lobourg Channel seems likely related to high magnitude flows that were derived from significantly upstream of the Dover Strait since the Lobourg Channel extends northeastwards into the Southern North Sea. We would then require a model of these flows breaking through the chalk barrier without any form of lake overspill. Of course final breaching of the Strait with a weakened rock dam could have been through such a process but this is difficult to test. The continuity of form of the Lobourg Channel through the Strait suggests to us that the

breach had already occurred.

The infill of the Fosse was also possibly over shorter durations - how many episodes, of scouring and filling, and how many glacial-interglacial cycles since the Elsterian... suggests shorter duration of infill.

Without chronology it is really difficult and dangerous to suggest links between scour and fill cycles and glacial-interglacial cycles. We prefer to keep things simple at this stage.

6. Fosse mapped surface: I didn't note in the discussion or methods whether there was evidence for the Fosse being on a single composite erosion surface, or within the constraints of the data, could their formation be multiple ages?

As far as we can tell the basal erosion surface is a single erosion surface. We do not see any evidence of cross-cutting relationships.

7. Internal surfaces: hard to see from the data presented the orientation of the scoop. Are these scour forms with steep headwall and shallowing and flaring down-dip? These are interesting, as difficult to envisage flows undergoing a hydraulic jump within the confines of a plunge pool.

Agree that this is difficult to see in our figures. Not all the internal erosion surfaces have a scoop-shaped morphology. We have thus removed that from the text as it was unclear.

8. Fosse morphology: 'C' looks more elongate NE-SW, whereas A, B, E are more elongate about NW-SE. Hard to see from data on D and F, yet in the text you describe them as similar in orientation a shape. Feels like there is more to get at here with the distribution of different orientations and depths with respect to underlying bedrock and tectonic features.

Agreed. We have now added discussion of this to the text.

9. Timing for non-experts: For the non-experts in the (somewhat opaque) world of Quaternary chronostratigraphy that might be reading, worth putting in some approximate years BP (and include Anglian and Wolstonian) to help accessibility of the text.

This has been clarified in the paper. We use Marine Isotope stages and provide approximate ages and relationship to glacial stages.

10. Geology for non-experts: What is the bedrock? This is labelled on the figures, and most UK geologists will know what underlies the chalk, which is strangely the only rock mentioned. You need to be more explicit in the text that the plunge pools, and other erosional bedforms, have been cut into the Gault Clay, and sometimes the Lower Greensand where deep enough. This is important for accessibility, but more so to characterise the bedrock to help discussion of the process or formation and location. The Gault Clay is less resistant compared to the Lower Greensand. Other authors have shown the important controls the bedrock geology has, but only the chalk is mentioned here – needs more clarity.

A new section has been added to the paper to discuss the role of bedrock lithology – lines 220-229.

11. Other processes: The authors used terms like ‘the megaflood landscape’, which is misleading. Many authors have shown/argued, the landscape/seascape is a palimpsest, and is a physiographic record of a wide range of erosional and depositional processes. Even if there were one or several large flood events, there has been significant modification by other processes, fluvial, wave, tide.

Agreed that this is a palimpsest but we would argue that the large-scale preserved morphology is largely a result of megaflood processes but we have changed our wording.

12. Linear ridges: The linear ridges do look very like the basal surface of submarine landslides - so the flood event origin, with a high concentration flow, is possible. Without any other evidence for glacial activity in this section this is highly speculative. Therefore, I'm not keen on this final paragraph - provocative is fine, but too speculative. A stronger paper if they strengthen their assertions on the number and mechanisms of floods through more discussion earlier.

Removed discussion of glacial origin of lineations.

Reviewers' Comments:

Reviewer #2 (Remarks to the Author)

Rereview of 'Two-stage opening of the Dover Strait and the origin of 'island' Britain' by Gupta et al. (NCOMMS-16-16909A)

Summary:

It was a pleasure to read this significantly extended and improved manuscript. The authors responded to the first round of edits in a clear and constructive manner, and I am encouraged to see that many of the suggestions for improvements from the reviewers have been taken on board. The longer format and new material has permitted the required deeper and richer observations and discussion that better support the profound implications from this data-rich study: that there were (at least) two stages to the opening of the English Channel/La Manche and there was a long period when multiple large waterfalls were active sourced from a lake confined by a chalk ridge across the Straits of Dover. The quantification of the erosional features, and the integration of the bedrock stratigraphy and structure are important additions, and the discussion of geological control and the strongly time transgressive nature of the plunge pool development and escarpment retreat prior to breaching is far clearer and greatly improves the robustness of the study. There remain discussions to be had on the degree of catastrophism required to shape the submerged landscape that we see today, and new studies from the Scablands on this topic could be integrated, however this requires more sedimentological and geomorphic data. In summary, this paper will be widely read and cited, and is an ideal study for Nature Communications to publish, and should be accepted, subject to some minor suggestions below.

Issues to address:

1. Model continuum: There is an interesting debate ongoing in the literature as to the degree of catastrophism needed to shape the landscape outboard of proglacial lakes – megaflood versus many large floods. Indeed, I recommend casting the 'models' in the Introduction (lines 48-57) as a continuum between catastrophic and 'slow fluvial' as most studies sit between these end members (including this one and Mellett et al. (2013)). The data here certainly points to many large flows during episodes or phases of landscape degradation rather than two discrete catastrophic events. Most of the time the authors avoid 'event' (e.g., modify line 291 to episodes). Also, I suggest adding a citation to a recent paper by Larsen and Lamb (2016) (Progressive incision of the Channeled Scablands by outburst floods, *Nature*, 538, 229-232. 10.1038/nature19817) to support the continuum of models in progressive shaping of the landscape downstream of proglacial lakes.
2. Line 115-116: Seems an odd location for this statement.
3. Antiform versus anticline: Why antiform? Widely anticline in the literature, and there is no doubt that the rocks across the antiform structure are younging from the centre to the edge.
4. Sequence of events: This is far clearer, and a great addition (lines 303-312), although I was crying out for a simple stepped palaeogeographic reconstruction, I understand why the authors avoided this for reasons of space, and robust chronology - something for the future.
5. Lobourg and Fosse age relations: Much clearer in text and figures. Suggest you delete line 205 as repeated on lines 217-220 with citation to figure.
6. Methods: Much improved in the methodology and data constraints meaning that the study can be deemed repeatable with the level of material here now.
7. Figure 2: Suggest moving the cross-section to below the map.
8. Figure 6: I found the red writing hard to read. Expand on alphanumeric code in the citation.
9. Figure 7: Likely need to increase font size for publication.

Reviewer #2 (Remarks to the Author):

Rereview of 'Two-stage opening of the Dover Strait and the origin of 'island' Britain' by Gupta et al. (NCOMMS-16-16909A)

Summary:

It was a pleasure to read this significantly extended and improved manuscript. The authors responded to the first round of edits in a clear and constructive manner, and I am encouraged to see that many of the suggestions for improvements from the reviewers have been taken on board. The longer format and new material has permitted the required deeper and richer observations and discussion that better support the profound implications from this data-rich study: that there were (at least) two stages to the opening of the English Channel/La Manche and there was a long period when multiple large waterfalls were active sourced from a lake confined by a chalk ridge across the Straits of Dover. The quantification of the erosional features, and the integration of the bedrock stratigraphy and structure are important additions, and the discussion of geological control and the strongly time transgressive nature of the plunge pool development and escarpment retreat prior to breaching is far clearer and greatly improves the robustness of the study. There remain discussions to be had on the degree of catastrophism required to shape the submerged landscape that we see today, and new studies from the Scablands on this topic could be integrated, however this requires more sedimentological and geomorphic data. In summary, this paper will be widely read and cited, and is an ideal study for Nature Communications to publish, and should be accepted, subject to some minor suggestions below.

Thank you. We are very grateful to reviewer 2 for their comments that have helped clarify our arguments. The longer format of Nature Communications has allowed us to present a more fully developed description of data and discussion.

Issues to address:

1. Model continuum: There is an interesting debate ongoing in the literature as to the degree of catastrophism needed to shape the landscape outboard of proglacial lakes – megaflood versus many large floods. Indeed, I recommend casting the 'models' in the Introduction (lines 48-57) as a continuum between catastrophic and 'slow fluvial' as most studies sit between these end members (including this one and Mellett et al. (2013). The data here certainly points to many large flows during episodes or phases of landscape degradation rather than two discrete catastrophic events. Most of the time the authors avoid 'event' (e.g., modify line 291 to episodes). Also, I suggest adding a citation to a recent paper by Larsen and Lamb (2016) (Progressive incision of the Channeled Scablands by outburst floods, Nature, 538, 229-232. 10.1038/nature19817) to support the continuum of models in progressive shaping of the landscape downstream of proglacial lakes.

Indeed, I recommend casting the ‘models’ in the Introduction (lines 48-57) as a continuum between catastrophic and ‘slow fluvial’ as most studies sit between these end members (including this one and Mellett et al. (2013).

We have added a sentence (line 63-67) in response to the above suggestion: “Thus the models for erosion of the palaeovalley networks downstream of the Dover Strait range from those proposing erosion by high-magnitude events to those suggesting relatively slow fluvial erosion.” However, we are hesitant to state that these form a continuum as this is suggestive of similarity of process, which is not correct.

Nevertheless in the discussion about the erosion of the Lobour Channel we have added ‘series of events’ to the following sentence: “The combination of seismic reflection data and high-resolution bathymetry data reveals that a second event (or series of events) was needed to fully open the Dover Strait. (lines 333-335)

“e.g., modify line 291 to episodes)”

Done

Also, I suggest adding a citation to a recent paper by Larsen and Lamb (2016) (Progressive incision of the Channeled Scablands by outburst floods, Nature, 538, 229-232. 10.1038/nature19817) to support the continuum of models in progressive shaping of the landscape downstream of proglacial lakes.

The paper by Larsen and Lamb does not suggest a continuum of models but proposes that the scale of outburst floods in the Channeled Scabland has been overstated. Since we do not discuss the scale of the floods in quantitative terms (the issues raised by Larsen and Lamb have long been known to the community) it seems unnecessary to cite the paper. Indeed, Larsen and Lamb state “The outburst floods that carved the Channeled Scablands were extraordinary under either end-member model” so they do not propose a continuum of models from slow fluvial erosion to megafloods, but rather models that range between the scale of the outburst floods. We have made this clear in the Discussion.

2. Line 115-116: Seems an odd location for this statement.

This has now been moved to earlier in the text.

3. Antiform versus anticline: Why antiform? Widely anticline in the literature, and there is no doubt that the rocks across the antiform structure are younging from the centre to the edge.

Changed to anticline

4. Sequence of events: This is far clearer, and a great addition (lines 303-312), although I was crying out for a simple stepped palaeogeographic reconstruction, I understand why the authors avoided this for reasons of space, and robust chronology - something for the future.

5. Lobourg and Fosse age relations: Much clearer in text and figures. Suggest you delete line 205 as repeated on lines 217-220 with citation to figure.

Done

6. Methods: Much improved in the methodology and data constraints meaning that the study can be deemed repeatable with the level of material here now.

Thank you for this comment.

7. Figure 2: Suggest moving the cross-section to below the map.

Done

8. Figure 6: I found the red writing hard to read. Expand on alphanumeric code in the citation.

Removed red writing.

9. Figure 7: Likely need to increase font size for publication.

Done